



# Inter-annual snow accumulation and meter-scale variability from trench measurements at Dome C, Antarctica

Adrien Ooms[1], Mathieu Casado[1], Ghislain Picard[2], Laurent Arnaud[2], Maria Hörhold[3], Andrea Spolaor[4], Rita Traversi[4], Joel Savarino[2], Patrick Ginot[2], Pete Akers[5], Birthe Twarloh[3], and Valérie Masson-Delmotte[1]

[1]Laboratoire des Sciences du Climat et de l'Environnement, LSCE/IPSL, CEA-CNRS-UVSQ, Université Paris-Saclay, Gif sur Yvette, France
[2]Institut des Geosciences de l'Environnement (IGE), Université Grenoble Alpes / CNRS, UMR 5001, Grenoble, France
[3]Department of Geosciences, Alfred-Wegener-Institut Helmholtz-Zentrum für Polar- und Meeresforschung, Bremerhaven, Germany
[4]Institute of Polar Sciences, National Research Council of Italy (ISP-CNR), Venice, Italy
[5]Discipline of Geography, School of Natural Sciences, Trinity College Dublin, Dublin, Ireland

**Correspondence:** Adrien Ooms (adrien.ooms@lsce.ipsl.fr) and Mathieu Casado (mathieu.casado@lsce.ipsl.fr)

**Abstract.**

The central regions of the East Antarctic ice sheet contains some of the oldest ice on earth, due to low snow accumulation rates and consistently cold conditions. One consequence of low accumulation however is that the little snow amount which is deposited at irregular times is more strongly affected by erosion and re-deposition by wind, inducing local mixing and

loss of snowfall events in the snowpack. This discontinuous deposition leads to highly disturbed snow layers, limiting the interpretation of climate records from ice cores in these regions to time scales larger than decades. In order to interpret climate records at higher temporal resolution it is crucial to assess and quantify the local patterns of snow accumulation leading to stratigraphic noise.

Here, we reconstruct the spatial and temporal variability of snow accumulation in Central Antarctica using chemical com-

position and physical properties in 35 vertical profiles sampled in a 50 m long snow trench at Dome C. We show that a high resolution alignment of the chemistry profiles is a suitable method for inter-annual dating of the single trench profiles, allowing the reconstruction of accumulation time series with a 1 year resolution over the last 20 years. This reconstruction shows annually-varying past surface configurations, with about 10 % of the surface subject to accumulation hiatus. More persistent patterns with timescales of years are also evidenced, causing difference in snow age of up to 4 years at similar depth in

neighboring profiles, highlighting the complex dynamics of the snow accumulation in central Antarctica.



## 1 Introduction

Antarctica is a key region for the global hydrological cycle due to its large reserve of fresh water, stored as ice sheets of up to 4 km in thickness (Frémand et al., 2023). Under the influence of climate change, the Antarctic ice sheet is losing mass at an overall rate of 127 Gt per year (Diener et al., 2021), and could contribute to sea level rise by up to 30 cm by the year
2100 (Intergovernmental Panel On Climate Change (IPCC), 2023). The magnitude and rate of the future mass loss remains associated with deep uncertainty (Intergovernmental Panel On Climate Change (IPCC), 2023), in particular the positive input coming from the atmosphere, referred to as the surface mass balance (SMB). In Antarctica, SMB is the net accumulation of snow and ice resulting from total precipitation, net sublimation and snow drift to the ocean. While mass loss is increasing due to the warming climate, it is partially offsetted by the projected increase in SMB (Hanna et al., 2024). However, it is difficult
to evaluate the change of SMB in Antarctica (Ekaykin et al., 2024), in part due to the difficulty to measure accumulation rates (Magand et al., 2007), as well as the large spatial variability at scales ranging from meters (Picard et al., 2019; Hirsch et al., 2023) to kilometers (Genthon et al., 2016; Dallmayr, 2025). This local spatial variability of accumulation is due to interactions between the freshly deposited snow, and wind scouring and redeposition (Sommer et al., 2018), an ensemble of processes often referred to as *stratigraphic noise*. Stratigraphic processes are key elements of the SMB (Zuhr et al., 2021), leading to patchy
deposition (Picard et al., 2019; Goodwin, 1990), appearing as self organized dunes, with shapes similar to those found in sand deserts (Poizat et al., 2024). Improvements of SMB modeling in regional climate models (Agosta et al., 2019; Wang et al., 2016; Mottram et al., 2021; van Wessem et al., 2018) requires better estimates of not only the mean SMB but also of the local spatial variability.

Several direct measurement methods are available to study accumulation and its spatial variability at the local scale, including
laser scanners, photogrammetry and stake networks, with temporal resolution of hours to multiple decades (Picard et al., 2019; Genthon et al., 2016; Ekaykin et al., 2023; Zuhr et al., 2021; Sommer et al., 2018). Chemical and physical tracers can also provide stratigraphic markers in the snowpack reaching several decades in the past (Caiazzo et al., 2021; Petit et al., 1982), but their temporal resolution strongly depends on accumulation rates. Snow pit chemical stratigraphy studies can reach annual resolution at sites with relatively high accumulation rates ($> 30$ cm yr$^{-1}$), such as sites in Greenland (Kjær et al., 2021),
and sites of moderate accumulation rates in Antarctica ($> 10$ cm yr$^{-1}$) such as South Pole (Dibb and Whitlow, 1996) and Drowning Maud Land (DML) (Hirsch et al., 2023; Münch et al., 2016). At DML, the implementation of a 2D sampling of the snowpack in the form of a trench (e.g. simultaneous sampling of snow profiles along a well defined transect), has shown to overcome the problem of stratigraphic noise and opened the possibility to derive information on deposition history of single snow layers (Laepple et al., 2016; Münch et al., 2016). Furthermore, the link between trace elements and snow micro-structure
seasonal variability has been studied (Moser et al., 2020). Studies of snow pits from Dome F (Hoshina et al., 2014; Iizuka et al., 2004), which features low snow accumulation similar to Dome C, successfully applied layer counting at 2 cm based on post-depositional seasonality of $SO_4^{2-}$.

In the Dome C snowpack, seasonal differences in snow cannot be resolved and therefore traditional layer counting in the physical or chemical stratigraphy is not possible (Caiazzo et al., 2021). Measurements of the chemical composition in snow



pits show multi-annual variability. Various sources of have been identified, including biological activity, sea salt, terrestrial

dust, and anthropogenic activities (Delmas et al., 1985; Cosme et al., 2005; Caiazzo et al., 2021). On this multi-annual signal

are superimposed sharp exceptional markers at multi-decadal to centennial intervals, due to volcanic or nuclear test fallout.

These markers serve as temporal tie points in the snow pack (Le Meur et al., 2018; Gautier et al., 2016; Petit et al., 1982), but

they only offer a decadal resolution, at best. In between these tie points, dating relies on linear interpolation which overlooks

the strong inter-annual variability of snow accumulation rate (Genthon et al., 2016). Estimates of snow accumulation rates

from the snowpack at Dome C at sub-decadal timescales are still lacking. One of the main limitations in using chemical and

physical tracers in snowpits as stratigraphic markers is their high spatial variability due to stratigraphic noise (Gautier et al.,

2016), which therefore requires several simultaneous snowpits to be captured.

In this work, we study the local spatial variability of accumulation rates at Dome C, using a new dataset of snow properties

measured in a 50 m long, 1.5 m deep snow trench dug in the 2019-2020 summer season, completed with similar measurements

in older snow pits dug in the 2000-2020 period (Traversi et al., 2009; Caiazzo et al., 2021; Gautier et al., 2016; Bertinetti et al.,

2020). The trench contains 35 snow profiles sampled at 3 cm vertical resolution, with horizontal inter-profile spacing ranging

from 30 cm to 2 m. Chemical tracers and snow microstructure (density, specific surface area (SSA) and penetrometry) were

measured. We observed that the measured sulfate and MSA concentrations represent a background signal that is recognizable

in the entire trench. Moreover, the sulfate signal in the trench can be matched with that of older snowpits sampled in the area

that overlap with the trench depth range. We combined these two observations to derive an age model for the entire trench.

Based on the age model, we derive a time series of accumulation rate at the sub-decadal timescale, and its spatial variability at

the decimeter to decameter scale. Finally, we compare our results to accumulation rates from atmospheric reanalysis and from

direct measurement methods.

## 2  Methods

### 2.1  Sampling and measurements

At Dome C (75° 06′ S, 123° 20′ E.), we excavated a 50 m long and 1.5 m deep straight snow trench in which 35 vertical snow

profiles were sampled over three weeks in December 2019. The trench was perpendicular to the main orientation of the wind

and the sampling was carried out on the wall face sheltered from the wind. Individual samples were collected directly from

the wall in 12 mL Corning tubes, and the 2D horizontal and vertical coordinates of the sample were recorded. We used a laser

level to establish a common reference height for all of the profiles, materialized by a horizontal string (visible in figure 1b and

d). Throughout the article, we refer to the absolute depth, $z$, of each sample as the vertical distance to this common height

reference, in contrast with the (regular) depth usually defined as the vertical distance to the local surface height. The surface

topography was rather flat (50 cm < $z$ <60 cm) in the left (Southward) section of the trench (0-35 m) and a 30 cm high dune

(minimum $z = 20$ cm) was present on the right (Northward) section (35-50 m) (Fig. 1a). Vertical profile sampling took place

every 2 horizontal meters along the whole trench and in addition finer horizontal resolutions (from 10 cm to 1 m) was applied

near the left end of the trench (between 0 m and 5 m), on the ascending slope of the dune (between 38 m and 44 m), and





on top of the dune near the right end of the trench (around 48 m). Along this trench, the profiles are labeled P followed by their horizontal position in meters measured from the left end of the trench. Most profiles were sampled with a 3 cm vertical

resolution and four profiles, P0, P0.3, P47.7 and P48, had a higher vertical resolution of 1.5 cm. For P0, when too little snow was available to measure chemical elements, we mixed them with high resolution samples of a neighboring profile, 30 cm to the left. Similarly, P0.1 was combined with samples 20 cm to the right, and P48 with samples 10 cm and 30 cm to the right.

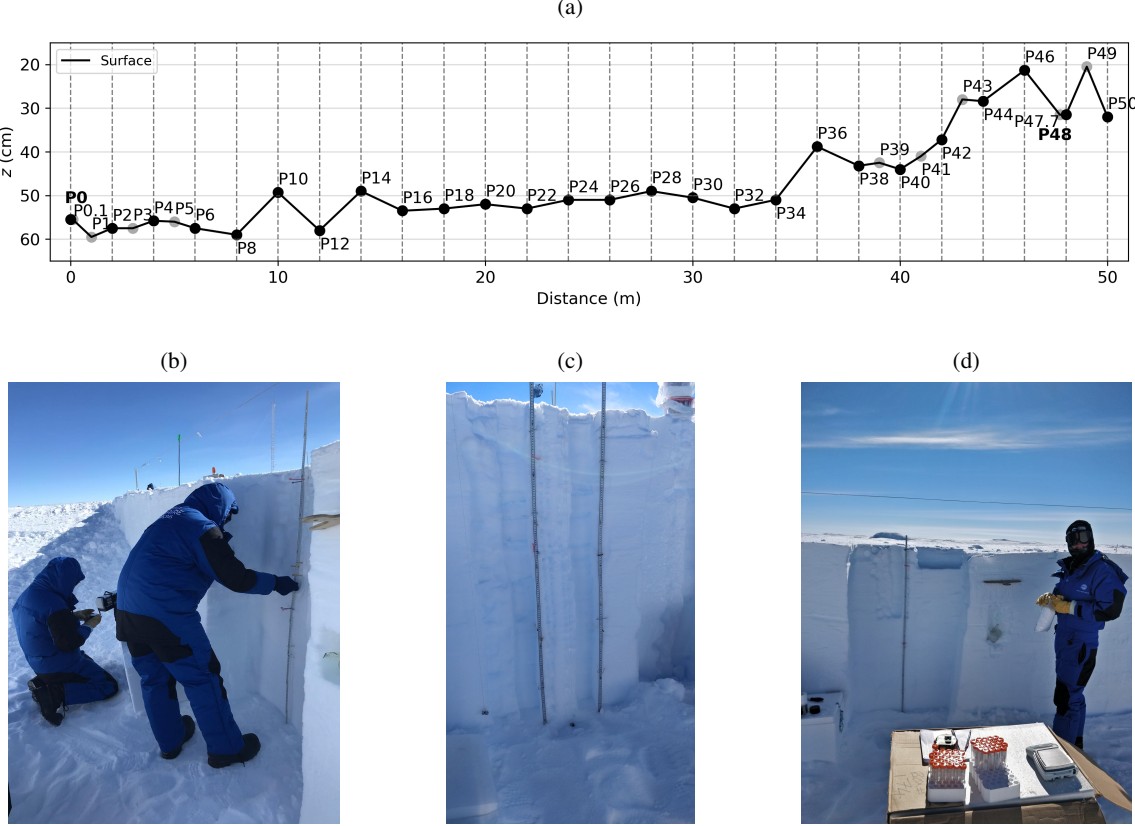

Figure 1: (a) Surface of the trench in absolute depth vertical coordinate and horizontal distribution of the profiles. Each profile is labeled as P + its horizontal distance to the profile P0. The vertical coordinate, absolute depth $z$, indicates the vertical distance to an arbitrary reference height. Profiles at 2 m spacing are displayed with black circles and constitute a subset of 26 profiles uniformly distributed along the trench. The surface is rather flat in the first 35 m, and there is a 30 cm high dune in the last 15 m. The majority of profiles is sampled at 3 cm vertical resolution. Higher resolution (1.5cm) profiles P0 and P48 are highlighted with bold labels. (b,c,d) Field photographs illustrating the trench sampling operations. The reference height $z = 0$ is materialized by the horizontal string (b,d).



After sampling, the snow samples in the Corning sampling tubes were parafilmed, and kept frozen during shipment to laboratories in Europe. The samples were analyzed by ion chromatography at the Alfred Wegener Institute (Bremehaven, Germany) and the Institut des Géosciences et de l'Environement (Grenoble, France) for the concentration of anions ($SO_4^{2-}$, $NO_2^-$, $NO_3^-$, MSA, $Cl^-$, $Br^-$, oxalate, $F^-$, acetate and formate) and cations ($Li^+$, $Na^+$, $NH_4^+$, $K^+$, $Mg^{2+}$, and $Ca^{2+}$).

We also deployed a Snow Micro Pen (Schneebeli et al., 1999) to estimate snow hardness. This measurement was carried out after the chemical sampling, with the probe applied 20 cm behind the exposed wall of the snow trench. Measurements were realized every 25 cm along the trench (200 profiles). The vertical resolution is extremely fine (0.004 mm) but the measurements were averaged at 1 cm resolution to obtain a macroscopic significance. The snow hardness probe was set to penetrate up to 150 cm, but it stopped earlier for several profiles due to too hard layers, leaving a few incomplete profiles. The Snow Specific Surface Area (SSA), a measure of the grain to pore interface surface area relative to the ice mass (unit: $m^2\ kg^{-1}$) was measured using ASSSAP (Arnaud et al., 2011), Libois et al. (2014). SSA was measured at 3 cm resolution along the same profiles as the chemistry samples. Snow density was measured at 3 cm resolution along the same profiles as the chemistry samples using a scale and squared metal sampler tool with a known volume.

The measurements carried out on each vertical profile were combined to create a 2-dimensional image for each chemical and physical tracer in the upper 1.5 m of snow (Sect. 3.1).

## 2.2 Inter-profile alignment

In order to derive accumulation from the snowpack, we constructed an age model for the trench. Instead of computing an age model for each profile independently, we proceed in two steps: in the alignment step (this section), we relate each depth profile in the trench to the depth of a reference profile by a squeeze and stretch function. The tie points for this transformation are obtained by matching features such as peaks and crests in the signal of three selected tracers: $SO_4^{2-}$, methanesulfonic acid (MSA), and SSA. In the dating step (next section), we compute an age model for a single reference profile. The age model for the entire trench is then obtained by applying the transformation of the alignment step to the reference age model.

To select tie points across the pits, we developed a versatile alignment software in Python hereafter referred to as *Alice* (*Al*ignment interface for *ice* cores). This software allows the user to select tie points across a variety of chemical and physical tracers. We used the three aforementioned tracers, with $SO_4^{2-}$ chosen as the primary tracer. Sulfate $SO_4^{2-}$ is considered as a being preserved in snow (Wagnon et al., 1999; Traversi et al., 2009), showing only moderate peak broadening except over timescales of thousands of years, due to diffusion (Barnes et al., 2003). Using *Alice*, a profile to be aligned is viewed against a previously selected reference profile. As the trench was sampled at the beginning of summer, before significant metamorphism had occurred (Picard et al., 2012), we used the high SSA values (>40 $m^2kg^{-1}$) of recently deposited winter snow (Libois et al., 2015) to match the topmost portions of the profiles and to detect hiatus in deposition. The rest of the profile is aligned by matching sulfate peaks, first with the largest peaks and then refined with sub-features. If the sulfate peak matching is ambiguous, MSA profiles were consulted. We note that MSA is well preserved in the first 100 cm depth, despite MSA disappearing at higher depth (Curran et al., 2002; Traversi et al., 2009).





We chose P0 as the reference profile because of its highest resolution and the presence of several sulfate features that are suitable as an alignment target. Despite these advantages, we observed that P0 features a lower SSA in its top snow than other profiles, and that on some profiles up to 3 cm of upper snow could not be matched to this reference. We interpret these facts as a missing snow layer at the top of P0. To solve this issue, we completed the 3 cm upper part of P0 with the mean surface
snow properties in the 3 cm upper part of all profiles over the trench. This permits us to align all profiles with respect to the reference up to the surface (Supplements, Sect. S2.1).

To summarize, the alignment routine goes as follows: **(1)** Display a profile to align P on top of the reference profile R leveling the top of the profiles; **(2)** looking at SSA values, evaluate whether top of the profile P contains fresh snow, or on the contrary is subject to an annual hiatus, and choose the topmost tie point (Fig. 2a and d); **(3)** align the rest of the peaks
with sulfate, starting with main crests and peaks, and refining with sub-features (Fig. 2b and e). **(4)** in case of ambiguous peak matching, switch to MSA profiles to resolve uncertainties (Fig. 2c and f).

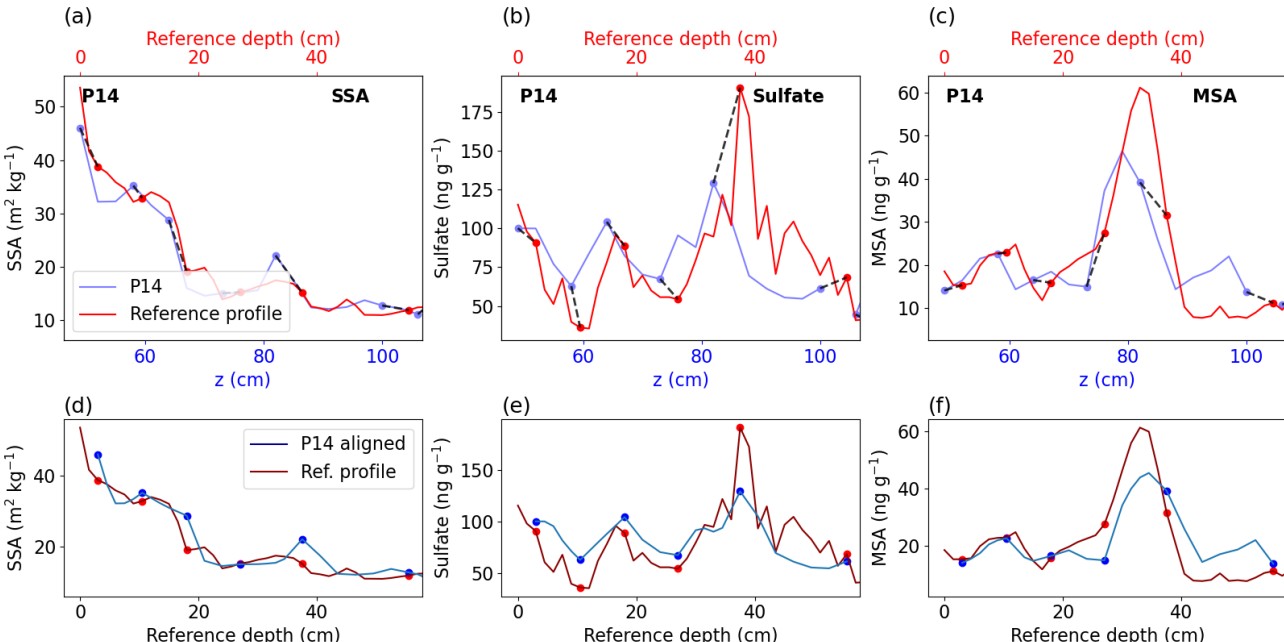

Figure 2: Example of alignment in the software *Alice*, showing SSA (a,d), sulfate (b,e) and MSA (c,f). The alignment of the topmost (50 cm) section of P14 is shown, illustrating the complementarity of the three tracers. In the upper panels (a,b,c), SSA, sulfate and MSA are shown side by side for P14 (blue) and reference profile (red) (alignment step (1)). The tie points are selected, and a preview of the aligned profiles are shown in the lower panel (d,e,f): the mid-range SSA value of 45 $\mathrm{m^2 kg^{-1}}$ on the first data point of P14 suggests that matching the head of P14 onto the second data point of the reference profile (alignment step (2)); Peaks of sulfate and MSA both support that we should match the rest of P14 slightly further right on the reference profile (alignment step (3)); the sharper drop of MSA around $z = 90$ cm refines the alignment on the slightly broader sulfate peak (alignment step (4)).





### 2.3 Dating of the reference profile

We construct an age model for the reference profile. This age model is then propagated to the aligned profiles in the entire trench. Based on previously published snow accumulation rates ($8 \, \mathrm{cm \, yr^{-1}}$, Genthon et al. (2016)), we did not expect to reach

the current depth of the Pinatubo sulfate level (about 2 m depth), deposited in Antarctica in 1992 (Cole-Dai and Mosley-Thompson, 1999; Caiazzo et al., 2021), so we have no volcanic horizon in the trench to serve as a dating point. Instead, our age model relies on similarities between the trench reference sulfate profile and sulfate concentrations in previous snowpits dug in the Dome C area in the 2000-2020 period. We were able to align these sulfate profiles together (Fig. 3) and considered that the top of each of these snowpits is an certain absolute dating point on the reference profile.

From the set of 22 snow pits dug at Dome C in the last 20 years (Gautier et al., 2016; Traversi et al., 2009; Caiazzo et al., 2021) we selected 13 that could be aligned with the reference profile of the trench (the others showed no remarkable feature in common with the reference in their upper 15 cm section). To construct the age model of the reference profile, we chose to only keep the top of each snow pit as an absolute tie point, as it is the date known with the highest confidence. The alignment was realized for the 13 profiles, giving 13 dating points on the reference profile. However only 6 dating points were kept for the

age model due to several snowpits being dug within the same month, resulting in duplicate tie points (Table 1). Error bars were assigned to the tie points depth on the reference profile. The error ranges between 5 and 10 cm depending on the quality of the match in sulfate signals. Small errors (5 cm) are assigned to the tie points just before a large and recognizable feature such as the broad sulfate peak in the 90-120 cm depth (December 2005), but conservative errors (10 cm) are assigned to tie points in a series of similar peaks, or in a section with little overlap, such as at depth 140-150 cm, corresponding to December 2000. For

a comprehensive use of all the data, we ensured that the error bars associated with the dating capture the variability observed across these duplicates. We used a different approach to obtain a dating point for the bottom of the reference, where alignment with an older snow pit is not possible due to the lack of overlap. The snowpits covering the Pinatubo horizon were associated with a linear age model between the date of sampling and 1992. This age model was then transferred to the reference after alignment (Fig. 3 (b) and (d), upper x-axis). The mean of these age models is used to get a temporal tie point for the bottom

of the reference profile. The error bar in dating for this tie point is the standard deviation of tie point age among age models. Figures for the alignment of all snow pits, including duplicates, are presented in the Appendix (Fig. A1).

In the dating routine, for visual aid, the snow pit to be aligned and the trench reference profile have a depth offset corresponding to the accumulated snow between times of sampling. The depth offset was calculated using 8 cm snow per year (Genthon et al., 2016).



| Sample date | Type | Max depth | Resolution | Species | Short name & reference | Pinatubo |
|---|---|---|---|---|---|---|
| 12-2000 | snow pit | 7.3m | 2.5 cm | $SO_4^{2-}$, MSA | Traversi2000 (Traversi et al., 2009) | x |
| 12-2005 | snow pit | 4m | 3 cm | $SO_4^{2-}$, MSA | Traversi2005 (Traversi et al., 2009) | x |
| 1-2008 | snow pit | 3.8m | 5 cm | $SO_4^{2-}$, MSA | Savarino2008 | x |
| | | | | | (Savarino, private communication) | |
| 1-2012 | ice core | 100 m | 2-3 cm | $SO_4^{2-}$ | VolSol (Gautier et al., 2016) | x |
| 12-2017 | snow pit | 3.8 m | 3 cm | $SO_4^{2-}$, MSA | Spolaor2017 (Spolaor et al., 2021) | x |
| 12-2017 | snow pit | 3.9 m | 3-4 cm | $SO_4^{2-}$, MSA | Traversi2017 (Bertinetti et al., 2020) | x |
| 1-2019 | snow pit | 0.5 m | 2 cm | $SO_4^{2-}$ | DCE_IGE15 | |
| | | | | | (Savarino, private communication) | |

Table 1: Summary information for the snow pits and ice cores used in the dating. Last column indicates whether the Pinatubo volcanic horizon of 1992 can be identified in the sulfate signal, allowing us to compute a linear age model for that snow pit.

## 2.4 Additional accumulation data

The accumulation rates we calculated in this study are compared to other available datasets. First, the changes of elevation measured every year around Dome C since 2005 (Genthon et al., 2016) in the GLACIOCLIM stake network which can be converted into accumulation. This network consists of 50 stakes located 2 km south to the Concordia station (75° 12′ S, 123° 32′ E.). The stakes are spaced approximately 40 m apart, arranged in a 1 km x 1 km cross. It is measured every year. Second, we compare our results with 3 years of high resolution accumulation data estimated from the elevation maps measured by an autonomous surface laser scanner called Rugged LaserScan (RLS), which was operational from January 2015 to December 2015 (period 1) and from January 2016 to December 2017 (period 2) (Picard et al., 2016a, 2019). The timeseries has daily resolution. The spatial setup of the instrument between both periods was different (total scanned area of 40 and 110 m² and horizontal resolution of 2 and 3 cm respectively). We only use the second period for detailed spatial analysis as its spatial extent is closer to that of the trench, and the continuous 2 years of data are necessary to capture important features of the accumulation dynamics.

The elevation change obtained from both the stake network and the laser scanner is converted into accumulation rate in mass equivalent using constant density value. Following the literature, we use a reference surface density of 320 kg m⁻³ (Genthon et al., 2016; Libois et al., 2015; Stefanini et al., 2025). In addition, we present the results obtained with a density of 295 kg m⁻³, which seem to be better suited according to our observations. This latter value is based on the trench density profiles (Fig. A2) and on other published data reporting density around 300 kg m⁻³ (Brun et al., 2011; Champollion et al., 2019).

We also compared our results with the outputs of the fifth generation ECMWF Reanalysis (ERA5) (Hersbach et al., 2020) estimating the accumulation as the difference between the total precipitation variable (mm w.e.) and the total evaporation and



summed over a year. The output of the reanalyses is selected at the grid point nearest Dome C (75° 00′ S, 123° 25′ E.). ERA5
180    accumulation rates shows good agreement with observations in Antarctica at the regional scale (Wang et al., 2025).



Figure 3



Figure 3: Example of dating in the software *Alice*, showing the alignment of the reference profile with a snow pit from 2017 ("Spolaor2017") (Spolaor et al., 2021) (a,b) and snowpit from 2005 ("Traversi2005") (Traversi et al., 2009) (c,d). On the upper panels (a,c), the snow pit profile (blue) is overlaid on top of reference profile (red). An initial depth shift is applied between both profiles, corresponding to the time difference between the two sample dates multiplied by a constant elevation change rate of 8 cm snow per year. Both of these profiles show the Pinatubo eruption horizon (Spolaor2021, depth = 210 cm, Traversi2005, depth = 115 cm), assigned with a deposition date in mid-1992. The lower panel (b,d) shows the result of the alignment, with the signals displayed on the reference profile depth scale. The snow pit surface tie point, associated to the sampling date of the snow pit, together with the Pinatubo horizon, provides an age model on the reference (upper $x-$axis, (b,d)). Only the top tie points (orange (b) and pink (d) squares) and the mean of the bottom tie points (orange (b) and pink (d) triangles) are kept for the final age model of the reference profile.

## 3 Results

### 3.1 2D profiles

Sulfate concentration shows a high profile-to-profile, and intra-profile variability (Fig. 4), characterized by a mean value of 72 ng g$^{-1}$ and a standard deviation of 20 ng g$^{-1}$. Global patterns of high and low concentrations are present across the trench, with small to high differences ($\pm$ 10-30 cm) in absolute depth. For instance, around $z = 80$ cm, sulfate concentrations above 100 ng g$^{-1}$ were measured in almost all profiles. Similarly, local minimum concentrations of about 50 ng g$^{-1}$ were measured in almost all profiles around $z = 140$ cm. The deepest coherent layer which can be identified in the aligned trench, corresponding to a minimum sulfate concentration of 50 ng g$^{-1}$, is located around $z = 180$ cm. There were about 15 measurements on the reference profile P0 outside of the $\pm$ 1 standard deviation range, which provided a convenient target for alignment with the other profiles. We refrained from interpreting these cycles as seasonal signal, and focused on identifying global multi-annual patterns between profiles. These patterns suggest that the chemical tracers can be used to obtain tie points between the depth profiles in the trench (Sect. 2.2). From these tie points, isochrones are drawn across the trench, as represented by the gray dashed lines in Fig. 4. We assume that these isochrones are associated with past accumulation events (Gautier et al., 2016), and therefore represent past configurations of the surface topography. An inspection of the density profile against depth shows that density is about 10% lower in the upper 6 cm section of the trench compared to the full profiles (295 kgm$^{-3}$ vs 328 kgm$^{-3}$), although no significant trend is found in the average density profile up to 1.5 m depth (Appendix, Fig. A2). This indicates that deformation of surface features is less than 10 % during the burial process. Note that the dating of the isochrones is carried out in Section 3.2.

The chemical tracer tie points are used to align the profiles together. The aligned trench shows an increase in inter-profile correlation ($\bar{r} = 0.3$) (Fig. S1) compared to the leveled trench ($\bar{r} = 0.1$) (Fig. A3), as expected since sulfate is used to align the trench. It is noteworthy that in horizontal averages, phase differences in the signal (phase noise) due to different deposition depths have been significantly reduced in the aligned trench compared to the leveled trench (Appendix, Fig. A3). The use of an





independent proxy is necessary to validate the alignment, which is provided by snow hardness measurements (Sect. 3.2). The horizontal decorrelation length of the sulfate signal (Appendix Sect. A3) is 1.05 m (e-fold) in the leveled trench and 1.26 m in
the aligned trench. This supports the use the evenly 2 m spaced out subset of profiles (Fig. 1 a, bold labels) when computing horizontal averages of tracer or accumulation rates, in order to avoid local scale correlation of the time series.



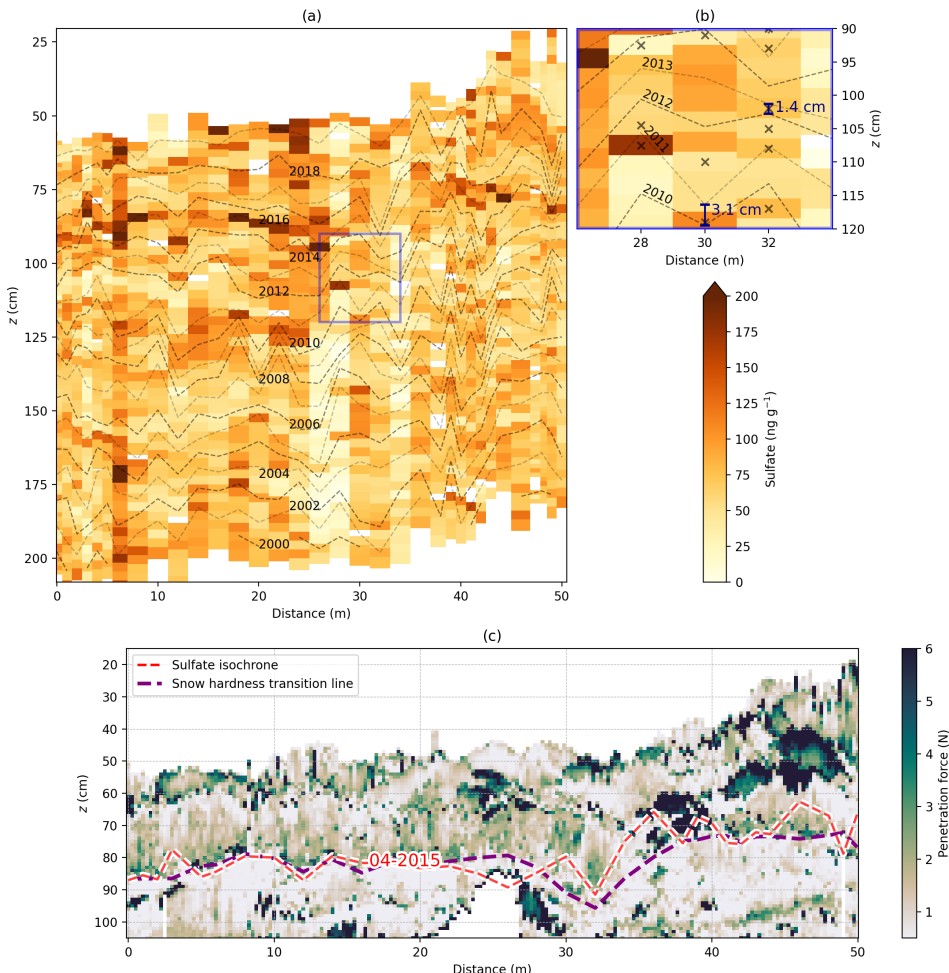

Figure 4: Dated stratigraphy obtained as a result of the alignment and dating procedure, overlaid on the sulfate concentration in the entire trench (a) and in a subset, chosen for its low accumulation (i.e. an accumulation hiatus) (b), and on the snow hardness (penetration force) in the upper section of the trench (c). Isochrones are interpreted as past surface configuration, and the depth between successive isochrones is the amount of snow accumulation in between these years. In (a), the blue rectangle indicates subset shown in (b). On (b) the zoomed in portion exhibits low accumulation (3.1 cm) at profile P30 for the year 2010, and accumulation below the 3 cm resolution threshold (1.4 cm) on P32 for the year 2012 (purple vertical segments). The tie points used for the alignment are indicated with light gray "x" signs. In (c), the dashed red line indicates the sulfate isochrone closest to a transition between soft and hard snow layers. Our age model attributes this past surface configuration to April 2015, which is contemporary to recorded extreme wind and temperature conditions of April 2015 described in Leduc-Leballeur et al. (2017) (red star, Fig. 5).





### 3.2 Absolute dating

Figure 5 shows the age model obtained on the P0 reference profile. The dating is calculated by linear extrapolation of 8 temporal tie points over the depth of the reference profile as detailed in Sect. 2.3.

The first dating point is on top 1.5 cm of the reference profile, which corresponds to the most recent snow deposition before the date of sampling (December 2019), with no error, since there is no accumulation hiatus at the surface in the reference profile (Sect. S2.1).

The next 6 dating points were obtained by alignment of the reference profile to snow pits dug in the Dome C area over the past 20 years. The depth uncertainty of dating points (horizontal error bars in Fig. 5) leads to an uncertainty in the snow age

at a given reference depth (vertical error bars) ranging from $\pm$ 0.5 years for the January 2019 tie point to $\pm$ 1.5 years for the January 2012 tie point.

The last tie point at the bottom of the reference profile is obtained from the mean over the 6 linear age models transferred from the aligned snowpits reaching the Pinatubo horizon (Table 1). The standard deviation among the age models gives an error estimate. They indicate that the snow layer buried at 151.5 cm depth was deposited in early 1999 $\pm$ 1 years. From the

surface snow deposited in June 2019, the trench thus archives about 21 years of snow accumulation, with a mean elevation increase of 7.3 cm yr$^{-1}$.



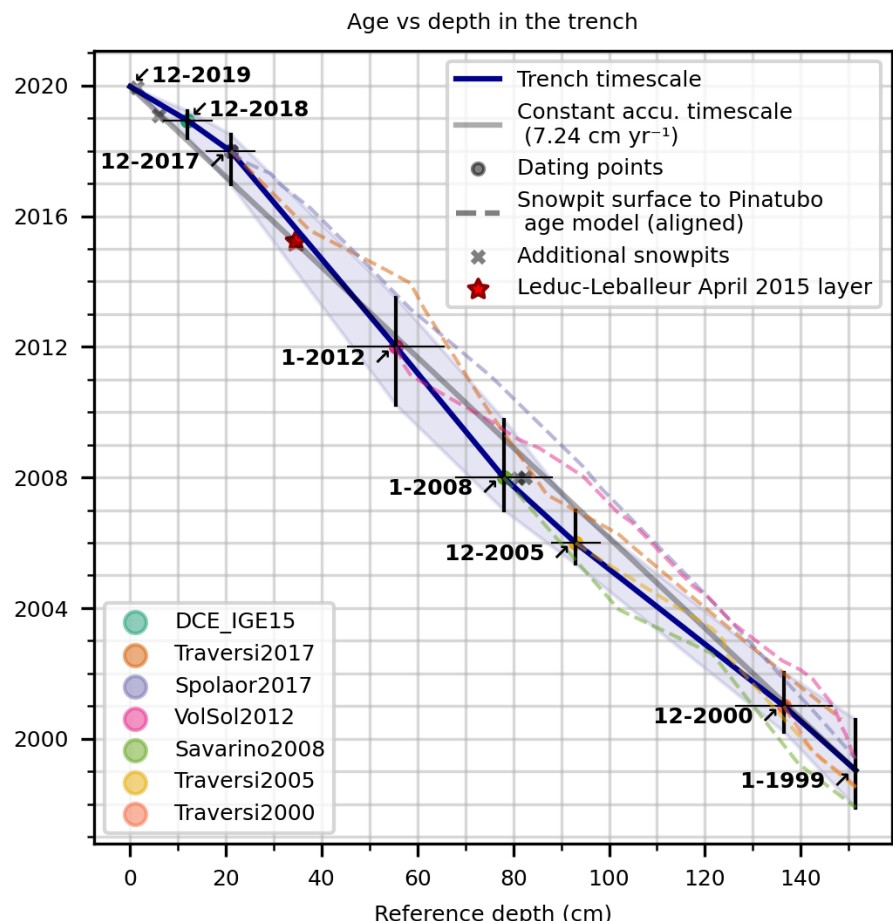

Figure 5: Age model on the reference profile: 8 dating points across the 151.5 cm depth of the reference profile; 1 dating point at the surface from the sampling time of the trench, 6 dating points in the bulk from the alignment with snow pits dug in the 2000-2020 period at Dome C (keeping only the tie point at the surface of the snow pit), and 1 dating point at the bottom of the trench by taking the mean of the 6 snowpits age models. The solid blue line shows the resulting age model for the reference used for the dated stratigraphy. Horizontal error bars show the depth uncertainty for the snow pit alignment while vertical ones show the corresponding age uncertainty. Dashed lines show the age models of the snow pits, obtained by linear interpolation between the surface and the Pinatubo volcanic horizon of mid-1992, and aligned onto the reference profile (Fig. A1b (2017, orange) and d (2005, pink), upper $x$-axis). The solid black line is the age model computed from the bottom 1999 tie point only, equivalent to a constant accumulation on the reference profile. The accumulation that would result from such a dating is provided for comparison in the Appendix (Sect. A4).





We evaluate how closely the dated isochrones in the trench (Fig. 4a) can be interpreted as past surface topography using the SMP as an independent variable not used for the alignment. Figure 4a shows dated isochrones drawn on top of the SMP measurements. We use the case of the mid-2015 isochrone as a validation of the past topography because it follows closely

a clear boundary between a softer and an harder layer detected by the SMP: the Pearson correlation coefficient between the hardness boundary and the isochrone is r = 0.67, p < 0.05, with only 4 cm average vertical distance between the two lines. The origin of the hard layer detected in mid 2015 is likely the signature in the snowpack of a remarkable meteorological event described in Leduc-Leballeur et al. (2017), as discussed in section 4.2.

### 3.3 Mean accumulation time series

In order to reconstruct accumulation, the stratigraphy based on the isochrones is combined with density measurement to convert observed elevation change in $\mathrm{cm\ yr^{-1}}$ of snow into accumulation in $\mathrm{kg\ m^{-2}yr^{-1}}$. The mean trench accumulation time series is calculated as the mean of the 26 profiles evenly distributed across the trench (2 m apart). The mean density measured over the trench is 328 $\mathrm{kg\ m^{-3}}$ with a standard deviation of 42 $\mathrm{kg\ m^{-3}}$. The surface snow (depth < 6 cm) has density of 293 $\pm$ 28 $\mathrm{kg\ m^{-3}}$, 10% lower than the average. Overall, we observe no detectable densification in the first 1.5 m depth, which

justifies that the distance between isochrones is representative of the accumulation depth at deposition time. The mean annual accumulation of the 26 trench profiles only differs by 4% on average when using mean density of 328 $\mathrm{kg\ m^{-3}}$ compared to full density profiles in the trench, while the difference in total accumulation for the period 2001-2019 are negligible (<0.1%) when using average vs spatially resolved densities (Sect. A7). Although the reference profile age model reaches as far as early 1999, the starting year for the accumulation reconstruction is set to 2001, since is it the first year identified on all of the profiles.

Figure 6 shows the resulting time series, which spans from early 2001 to late 2019 with a mean accumulation of 23.9 $\pm$ 1.5 $\mathrm{kg\ m^{-2}\ yr^{-1}}$. The error is estimated as the combination of the uncertainty of the depth of the 2001 layer (135 $\pm$ 3 cm, ) and the dating uncertainty around $\pm$ 1 year in 2001.

GLACIOCLIM stake network elevation change are converted to accumulation rate using a constant density value $\rho_{\mathrm{surf}}$. The reference density value is $\rho_{\mathrm{surf}}$ = 320 $\mathrm{kgm^{-3}}$. Based on the density observed in the surface snow in the trench, we propose

a second value of density of $\rho_{\mathrm{surf}}$ = 295 $\mathrm{kgm^{-3}}$ to convert snow elevation change to accumulation rate. With a density of 320 $\mathrm{kg\ m^{-3}}$, the accumulation rate estimate is 26.7 $\pm$ 0.3 $\mathrm{kg\ m^{-2}\ yr^{-1}}$ (for 2004-2019). A density of 295 $\mathrm{kg\ m^{-3}}$ yields 24.6 $\pm$ 0.3 $\mathrm{kg\ m^{-2}\ yr^{-1}}$ (for 2004-2019). The ERA5 reanalysis time series for the 2001-2019 period is 24.9 $\mathrm{kg\ m^{-2}\ yr^{-1}}$ precipitation and 2.6 $\mathrm{kg\ m^{-2}\ yr^{-1}}$ evaporation rate, resulting in 22.3 $\mathrm{kg\ m^{-2}yr^{-1}}$ net accumulation rate. We note that the net accumulation from ERA5 does not include potential contribution of snow drift, contrary to the trench and stakes data.

Turning to spatial variability, taking $2\,\sigma/\sqrt{n}$ as the spatial envelope where $\sigma$ is the standard deviation across profiles and $n$ is the number of profiles, we obtain $2\sigma/\sqrt{n}$ = 4.5 $\mathrm{kg\ m^{-2}\ yr^{-1}}$ for the trench (26 profiles, 50 m scale) and $2\sigma/\sqrt{n}$ = 6.5 $\mathrm{kg\ m^{-2}yr^{-1}}$ for GLACIOCLIM stakes (50 profiles, 1 km scale) ( 5.8 $\mathrm{kg\ m^{-2}yr^{-1}}$ with $\rho_{\mathrm{surf}}$ = 295 $\mathrm{kgm^{-3}}$) . Finally, the inter-annual standard deviation in accumulation rates is quite comparable between the three datasets, with 4.4 $\mathrm{kg\ m^{-2}\ yr^{-1}}$ for the trench (2001-2019 period), 5.7 $\mathrm{kg\ m^{-2}\ yr^{-1}}$ for GLACIOCLIM stakes with $\rho_{\mathrm{surf}}$ = 320 $\mathrm{kgm^{-3}}$ (2004-2019 period)

( 5.2 $\mathrm{kg\ m^{-2}yr^{-1}}$ with $\rho_{\mathrm{surf}}$ = 295 $\mathrm{kgm^{-3}}$) and 3.8 $\mathrm{kg\ m^{-2}\ yr^{-1}}$ for ERA5. Restricting to the 2004-2019 period does not



change the results by more than 3% for the trench and ERA5. While the amplitudes are also similar, there is a slight difference in phase (Fig. 6), which could be attributed to local spatial variability of the 50 m long trench. This point is discussed in the Appendix A5.





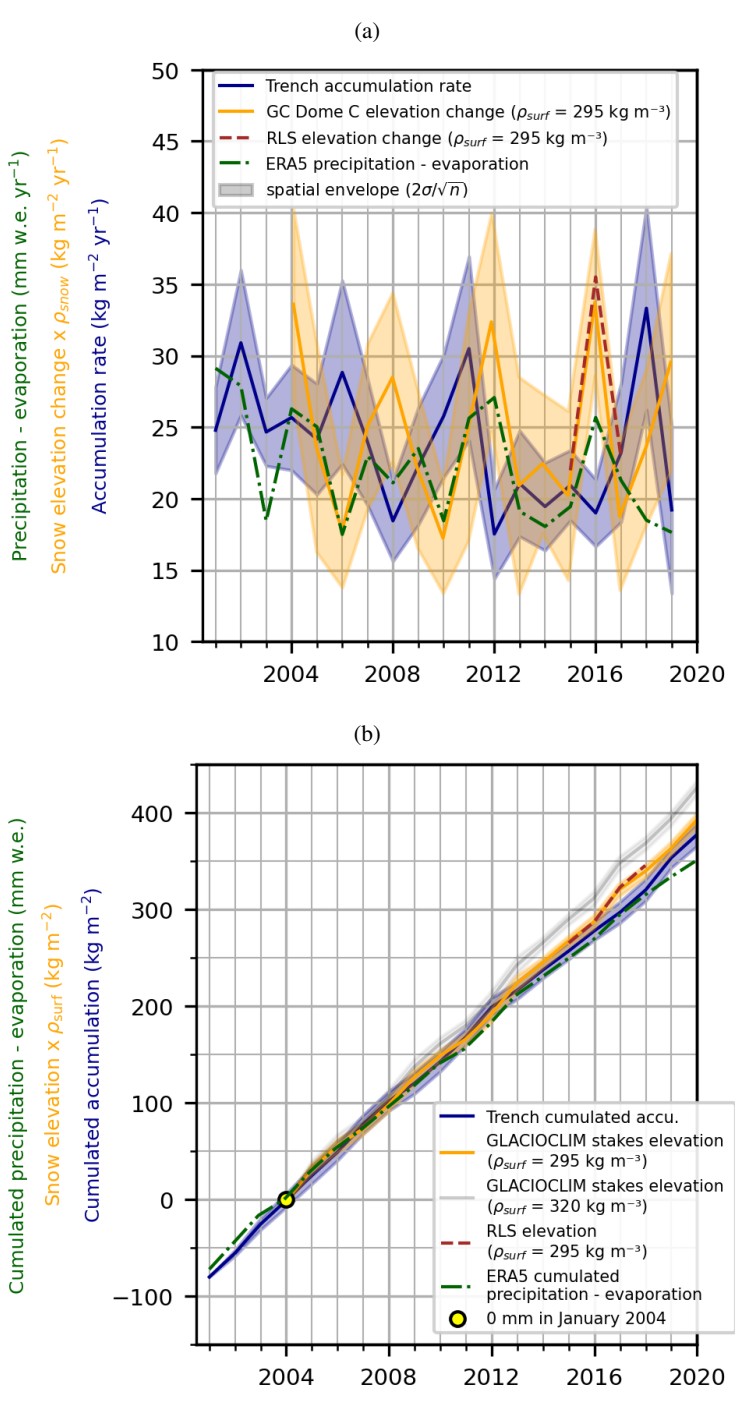

Figure 6



Figure 6: (a) Accumulation time series. GLACIOCLIM stakes and RLS data are converted to accumulation with the constant density value of $\rho_{\text{surf}} = 295 \ \text{kg m}^{-3}$. For comparison, we also show the accumulation rate of GLACIOCLIM stakes with a density of $\rho_{\text{surf}} = 320 \ \text{kg m}^{-3}$. The envelopes display twice the standard deviation in the spatial component, divided by the square root of the number of profiles (50 stakes for GLACIOCLIM, 26 profiles for the trench). ERA5 accumulation is the annual total precipitation minus evaporation. The mean values of accumulation are 23.9 $\text{kg m}^{-2}\text{yr}^{-1}$ (6% accurate) for the trench, 22.3 $\text{kg m}^{-2}\text{yr}^{-1}$ for ERA5 and 24.6 $\text{kg m}^{-2}\text{yr}^{-1}$ for GLACIOCLIM stakes (1% accurate). (b) Cumulated accumulation time series. Series have been set to an initial value of 0 mm in January 2004 (first GLACIOCLIM stakes measurements), and RLS initial value is set to match GLACIOCLIM in January 2015. Trench cumulated accumulation follows closely ERA5 net accumulation (precipitation - evaporation). $\rho_{\text{surf}} = 295 \ \text{kg m}^{-3}$ seems to give better agreement between trench and stakes data.

### 3.4 Frequency of accumulation hiatus

Fig. 4b shows that the 2012 and 2013 isochrones are in a very narrow range of depths ($< 3 \ \text{cm}$) around profile P32, indicating a low accumulation at this location for the year 2012. Such occurrences of accumulation below 3 cm are interpreted as near accumulation hiatus (compared to the 8 cm of annual snow accumulation), i.e. periods with negligible accumulation or even erosion at a given location. Considering that the vertical resolution of the alignment is 3 cm, i.e. the lowest depth difference that we can resolve, we refer to periods with smaller amount of snow accumulation as *hiatus*. The hiatus fraction for an 265 accumulation event is the fraction of profiles with a hiatus. The probability of hiatus for a time scale is the mean hiatus fraction for all accumulation events corresponding to this time scale. The frequency of hiatus should be close to 100% for a very short time difference, because almost all snow deposits are less than 3 cm, and gradually decreases with the considered time-scale.

Fig. 7 shows the probability of hiatus occurrence as a function of the time scale, up to 3 year periods, in the trench and in the RLS area. The 19 year record of accumulation in the trench, with 3 cm vertical resolution, equivalent to at least 6-month 270 temporal resolution, is compared to the 2 years time series of the RLS (period 2) with daily resolution. There are 26 evenly distributed accumulation time series 2 m apart in the trench, compared to 110 elevation change timeseries of 1 $\text{m}^2$ averages of the RLS (110 $\text{m}^2$ scanned area).

The hiatus probability is computed for the trench isochrones at monthly resolution for periods from 6 to 36 months. We did not consider periods shorter than 6 months for the trench as it is under our resolution limit. Hiatus probability reaches 42 % at 275 6 month (15% standard deviation across accumulation events), and 5 % at 1 year (5 % standard deviation). We found no hiatus (0%) extending over 2 years.

For the upper section of the trench (depth $< 20 \ \text{cm}$), a second computation of hiatus statistics is possible based on SSA values. SSA is decreasing for older snow and the SSA values corresponding to snow older than 1 year can be deduced from the dated trench. We found empirically a threshold value between 35 to 37 $\text{kg m}^2$ for 1 year old snow. The fraction of profiles 280 with SSA values under this threshold is 3/26 and 6/26 respectively, resulting in a annual hiatus probability of 10-20% for the year 2019, higher than the estimation with the chemistry dating. The details of this alternative approach is provided in the





Supplements (Sect. S6.2). The same computation with the 2 year SSA threshold value of 23 $\mathrm{m^2\ kg^{-1}}$ yields 0 profile with 2 year accumulation hiatus, agreeing with he chemistry dating.

These results are compared against the RLS dataset. The hiatus probability is computed for daily elevation maps for all
periods between 1 to 730 days. The results in the RLS area are similar with an agreement at 6 months, a marked decrease with only a slightly higher (non statistically significant) probability of hiatus in the range 6 - 18 months. No accumulation hiatus over 2 years or more are detected. There is 41 % mean hiatus probability at 6 month, with a 15% standard deviation across accumulation events, and a 10 % probability at 1 year, with a 10 % standard deviation. This is close to the $5 \pm 5$ % estimate in the trench, considering that the observation period is 19 years for the trench and 2 years for RLS. We note that there are 2
annual accumulation events within the 19 years of the trench record that show a 20% hiatus fraction.

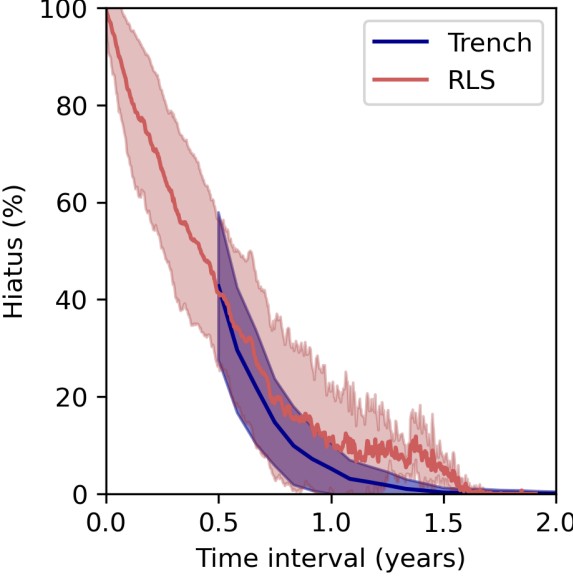

Figure 7: Probability of hiatus for a given time period in the Trench (blue) (2001-2019) and for RLS (red) (2016-2017). For a given time period, the probability of hiatus is the fraction of accumulation events with snow elevation change less than the threshold value of 3 cm (the resolution of the trench alignment). Periods shorter than 6 months are under the temporal resolution limit of the trench. The annual hiatus probability is $5 \pm 5$ % for the trench and $10\% \pm 10\%$ for the RLS. There are no accumulation hiatus extending beyond two years.

## 3.5  Accumulation and Topography

Isochrones in Figure 4a show that the snow accumulation does not lead to a flatter surface with time, but instead the persistence of several years of some topographic features over time is observed. For instance, the 10 cm high bump around P10 took 5 years before being erased. The larger 30 cm high dune located in the 35-50 m section of the trench seems to persist throughout
the accumulation record (about 20 years). We therefore hypothesize that accumulation and topography are relatively decoupled




at the meter scale. In order to quantify this phenomenon, we define the following measure of local topography. At a given point on an isochrone, we define $\Delta z$ as the difference between absolute depth $z$ of the isochrone at that point, and the mean absolute depth values of the two neighboring points (at +2 m and -2 m). $\Delta z$ measures the curvature, it is negative for bumps and positive for holes.

Figure 8 shows the distribution of snow accumulation against $\Delta z$. While the highest accumulation amounts ($> 10\,\mathrm{cm\,year^{-1}}$) are more often associated to holes as expected, the bulk of the distribution is very spread out. A linear regression gives a positive slope of 8 % $\mathrm{yr^{-1}}$, and a determination coefficient of $r^2 = 0.08$ (p-value $< 0.05$). This linear slope value of 15 % $\mathrm{yr^{-1}}$ is very low compared to 100 % $\mathrm{yr^{-1}}$, the value expected for the annual filling of holes. It means that a hole on the surface (resp. bump) receives only 15% more (resp. less) annual accumulation than its surroundings. The low determination coefficient also

shows that there is no consistent relationship between topography and accumulation. This supports the hypothesis of a strong multi-year persistence of the meter-scale topography.

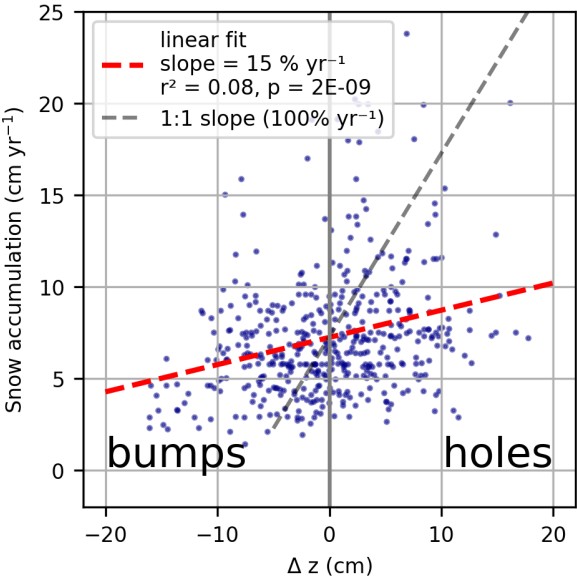

Figure 8: Correlation between annual snow accumulation and local topography anomaly $\Delta z$ for the trench annual isochrones interpreted as the past surface topography. $\Delta z$ measures the local curvature and is defined as the difference in absolute depth at a point compared to its direct neighbors ($\pm 2$ m ). Holes are on the positive side of the x-axis (depth greater than neighboring profiles), and bumps are on the negative side. The linear regression slope = 15 % $\mathrm{yr^{-1}}$ ($r^2 = 0.08$, p<0.05), red between accumulation and topography (holes accumulate more). The slope of 100 % $\mathrm{yr^{-1}}$ corresponds to annual filling of the holes.



## 4 Discussion

### 4.1 Limitations of the alignment

The dated stratigraphy constructed on the trench data by inter-profile alignment and dating of the reference profile was success-
ful in recovering sub-decadal characteristics of accumulation patterns at Dome C both in mean accumulation values, spatial
and temporal variability (Sect. 3.3). However, these accumulation estimates are constrained by limitations arising from the
uncertainties caused by the inter-profile alignment (Sect. 2.2), as well as the reference, single profile dating (Sect. 2.3).

One intrinsic limitation of the method is the number of recognizable features in the sulfate signal. There are about 15 peaks
or crests higher or lower than the $\pm 1\sigma$ range on P0, which is reflected in the fact that 14.5 tie points per profile were used
on average for the alignment, or one every 10 cm. Compared with the 21 years present in the cores, this suggests that there
is no exploitable traces of seasonal variability in the sulfate signal at Dome C. This was tested by sampling 4 profiles at 1.5
cm resolution instead of 3 cm, namely P0, P0.1, P47.7 and P48. The average number of tie points for the 3 high resolution
profiles beside the reference profile P0 was 13. This means that with 3 cm vertical resolution, we can effectively resolve the
stratigraphy exploiting the full significant sulfate signal, and that the related uncertainty is effectively the precision related to
sampling resolution.

The sampling resolution of 3 cm in the trench is to be compared to the annual mean accumulation of $7.8 \pm 1.4$ cm yr$^{-1}$
($23.9 \pm 4.4$ kg m$^{-2}$ yr$^{-1}$, Sect. 3.3), meaning our precision is 3/8 = 4.5 months on average and 3/6 = 6 months for the thinner
layers. To evaluate the impact on accumulation rate estimates, we have performed a sensitivity test on the alignment by shifting
isochrones by a random height of up to 3 cm around our final alignment. The result is an uncertainty of $\pm 0.8$ kg m$^{-2}$ yr$^{-1}$ on
accumulation, or approx 3% (Sect. A4).

In addition, we estimated the global uncertainty of the dating procedure of the trench (Appendix, Fig. A1) to roughly $\pm 1$
year, evenly affecting all profiles (Sect. 4.2). This was confirmed by the comparison between the dating of the trench and the
SMP hard layer detected in March 2015, and attributed to exceptional meteorological condition around a strong wind event.
The April 2015 sulfate isochrone is co-localized within 4 cm vertical distance with a boundary between higher and lower snow
hardness (Fig. 4c ). Leduc-Leballeur et al. (2017) documented a remarkable meteorological event in March 2015, when wind
exceeding 7 ms$^{-1}$ removed about 2 cm of fresh snow from the surface. This type of high wind speed would typically lead to
harden crust, therefore the 2015 snow hardness transition could be the signature of this event. Another independent validation
arises from the isotopic composition measurements where the maximum $\delta^{18}$O values identified in the trench in summer 2014
can be found in surface snow isotopic composition time series in January 2014 and minimum values of d-excess identified
in the trench in summer 2015 can be found in surface snow isotopic composition time series in January 2015 (Casado et al.,
2021). Overall, we conclude that the trench stratigraphy gives local accumulation with a 6 month inter-profile resolution, and
$\pm 1$ year global uncertainty.

Beyond the temporal resolution limits of this method, we also cannot obtain negative accumulation by studying the final
accumulation product in the snowpack. By contrast, the GLACIOCLIM stakes network exhibits negative annual accumulation
values for 10% of the stake measurements. While negative accumulation is out of reach with the trench approach, we inves-



tigated low accumulation years (near hiatus) and found similar statistics for low accumulation events (< 3 cm) between the trench and RLS datasets (Sect. 3.4). Here again, we show that our accumulation estimates are reliable when considering layers representing at least 6 month of accumulation.

## 4.2 Reference profile and limitation of dating

We have chosen to work with a reference profile as a target for the alignment of all the profiles in the trench, instead of pair by pair alignment. Beyond the obvious gain in time in the alignment ($n$ vs $n(n-1)/2$ pairs to align, where $n = 35$ is the number of profiles) and dating (only one profile to date), we also gain in coherence as all features recognizable on the reference were matched precisely. Reference profile alignment however comes with some caveats, and the dating of the reference remains the main source of uncertainty.

First, the reference profile can itself be subject to hiatus. In section 3.4, we found that accumulation hiatus occurs in 5% of the profiles on average. Comparing the sulfate records in five 100 m long ice cores, Gautier et al. (2016) found a higher probability of 30% of missing one single volcanic event in a single core. Accumulation hiatus on the reference profile lead to missing minima or maxima in its sulfate record. Such a missing feature would be present in 95% of the trench profiles and inaccurately aligned on the reference. However, with our approach, these effects are automatically corrected for on average.

For instance, if year 2015 was missing in the reference, but years 2014 and 2016 were clearly identified,the accumulation for 2014-2015 and 2015-2016 may be inaccurate, but the accumulation of the two years period 2014-2016 would still be correct.

Second, the uncertainty of dating of the reference profile must also be considered. When aligning the reference to older snow pits, we estimated a typical error of $\pm 5$ cm in the depth associated to the top of the snow pit which propagates into $\pm 1$ yr in the age depth scale of the stratigraphy (Sect. 3.2). We have performed a sensitivity test with variations in the age

model of the reference with $\pm 1$ yr variations, and found variations in accumulation as high as 5 kg m$^{-2}$ yr$^{-1}$, i.e. about 20% in individual annual accumulation rates. (Sect. A4). The uncertainty on the total accumulation of the 19 years record is 6% (Sect. 3.3). Previous studies of snow accumulation rates have used seasonal signal in chemical tracers, in particular sulfate, to resolve annual layers. This has been successfully applied to higher accumulation sites such as DML (Moser et al., 2020) as well as low accumulation sites such as Dome F (Iizuka et al., 2004; Hoshina et al., 2014). Layer counting is based on sulfate

peaks observed in austral summer due to the biogenic activity (Cosme et al., 2005). We have not tried this approach in the present study, as it is not applicable to our coarse resolution ( 3 cm ). The higher resolution (1.5 cm) profiles P0 and P48 do exhibit high frequency seesaw variability with very variable amplitude, but peak counting results in 15-25 peaks depending on counting strategy, and it is unclear whether it could be interpreted as annual layers.

To test the sensitivity of the reference profile, we performed another alignment and dating procedure with the profile P48,

and the results are presented in the Appendix. This alternative reference alters the accumulation reconstruction at the annual scale, with discrepancies of the same order as those resulting from the 20% dating uncertainty. The mean accumulation in the period 2001-2019 is within the error bars of the first alignment using P0, and the overall shape of the mean accumulation time series is also conserved (e.g. high value around 2012, low value around 2009, low values for 2014-2016, Fig. A4). Therefore,





we argue that, within conservative error values on the dating alone, our alignment method yields a robust and reproducible
accumulation reconstruction that is able to capture its inter-annual variability with good accuracy.

### 4.3   Comparison of accumulation time series

Different accumulation time series based on the trench and other lines of evidence (Sect. 3.3) differ slightly in mean values and
inter-annual variability, but overall agree within ranges of uncertainties.

The mean accumulation rates are $23.9 \pm 4.5$ kgm$^{-2}$yr$^{-1}$ (mean value for the period $\pm$ spatial envelopes) with a 6% un-
certainty for the trench (2001-2019) and 22.9 kgm$^{-2}$yr$^{-1}$ for ERA5 (3.3. The lower value of ERA5 is just within the 6%
uncertainty of the trench snow accumulation rate. The mean accumulation is $24.6 \pm 6.0$ kg m$^{-2}$yr$^{-1}$ (with a 1% uncertainty)
for GLACIOCLIM stakes (2004-2019) with the surface density $\rho_{surf} = 295$ kgm$^{-3}$, or $26.7 \pm 6.5$ kg m$^{-2}$yr$^{-1}$ with $\rho_{surf} =$
320 kgm$^{-3}$. The conversion of GLACIOCLIM stakes elevation to accumulation rate relies on the choice of a constant density,
traditionally set as $\rho_{surf} = 320$ kgm$^{-3}$ (Genthon et al., 2016; Libois et al., 2015), which considers a linear regression over a 5
385   m density profile. Other studies recommend a density for surface snow of 300 kgm$^{-3}$ instead (Brun et al., 2011; Champollion
et al., 2019). This is more in line with the direct measurements of surface density or of the value measured in the trench of
295 kgm$^{-3}$ (depth < 6 cm). This lower value of surface density removes the 10% gap that would otherwise arise between
GLACIOCLIM stakes and trench snow accumulation rates (Fig 6b). The reason for the overestimation of surface density could
be due to metamorphism of surface snow upon burial in the firn. Lighter snow only constitutes the superficial 5-10 cm layer
effectively considered by the stake readings, and a linear regression over more than a meter depth will over-estimate this lower
surface density. We note that the mean density in the trench in the depth range 0-1.5 m is 328 kgm$^{-3}$, and a linear regression on
the 1.5 m gives a surface density value of 320 kgm$^{-3}$ which agrees with the density profile of Genthon et al. (2016) (Appendix,
Fig. A2).

We note that Stefanini et al. (2025) found a mean elevation change of $7.3 \pm 0.2$ cm yr$^{-1}$ for the 2011-2023 period, using a
stake farm of 13 poles situated 800 m south-west of Dome C, which is closer to the trench site than the GLACIOCLIM stake
farm by about 1 km. This is equivalent to $23.4 \pm 0.6$ kg m$^{-2}$ yr$^{-1}$ with $\rho_{surf} = 320$ kgm$^{-3}$ and $21.5 \pm 0.6$ kg m$^{-2}$ yr$^{-1}$ with
$\rho_{surf} = 295$ kgm$^{-3}$. The mean accumulation rate in the trench is 22.4 kg m$^{-2}$ yr$^{-1}$ for the period (2011-2019), 6 % accurate,
which would be in agreement with the stake farm with either values of $\rho_{surf}$.

In addition, the trench accumulation rate lies within the spatial envelope of the GLACIOCLIM stakes network with either
value of $\rho_{surf}$ as the GLACIOCLIM stakes cover a 1 km x 1 km. Discrepancies due to decameter-scale variability of snow
deposition needs to be taken into account, and the 50-m long trench could be a specifically low accumulation area at Dome C.

We compare the amplitudes of mean annual accumulation time series between the trench and the GLACIOCLIM stakes.
The maximums displayed by the trench accumulation time series have an amplitude comparable to the maximum accumulation
annual values documented by direct measurement methods and reanalysis data. The peaks however are out of phase by $\pm$ 1
year for the year 2011, and $\pm$ 2 year for 2006 and 2008 (Fig. 6). This could be due to errors in the dating of the trench chemical
stratigraphy, which brings uncertainties of the order of $\pm$ 1 year. However, such dephasing of $\pm$ 1 year is observed within the
GLACIOCLIM network itself (for the 2009 or 2013 peaks for instance), when considering stakes at the opposite sides of the



1 km x 1 km cross (Fig. A5). Therefore we cannot rule out that the differences in phase are not due to the intrinsic spatial variability of accumulation at km scale, as the sites of the stake farm and the trench are approximately 2 km apart.

Overall, having multiple trenches at and around Dome C would be necessary to be able to separate the spatial variability at the different scales (decimetric to kilometric) which seems to affect snow accumulation.

### 4.4  Hiatus, dune and impact on ice-cores

Laepple et al. (2016) suggests that more than 90% of the isotopic signal archived in ice cores is noise, suggesting that it is necessary to average more than 10 ice cores 5 to 10 m apart is necessary to retrieve a coherent local isotopic signal in
interior sites in Antarctica (Münch et al., 2016), and minimize noise due to accumulation variability. We evaluate the impact of accumulation variability on sub-decadal reconstruction from ice-cores drilled at Dome C. We have showed that the 30 cm dune observed in the 35-50 m section of the trench is persistent over 20 years. Taking two profiles from the trench as an example of two ice-cores drilled several meters apart, sitting on top of different configurations of past dunes, we see that they would have a depth offset of up to 30 cm that propagates over more than a 20 year long section of the cores. We quantify the age difference
that snow samples at the same depth could have for this pair of hypothetical ice cores. Computing the distribution of years at a given depth based on our trench age model, the 98% quantile is 3.4 years. With our high resolution alignment, this mismatch is reduced to ± 6 month. (Sect. 4.2).

For multi-decadal or centennial records of longer ice-cores dug at Dome C, traditional alignment relies on major volcanic peaks horizons giving tie points every 50 years on average (Castellano et al., 2005). Our approach suggests that this volcanic
peak matching can be taken as a preliminary alignment, and be further refined within 50 year windows using background sulfate signal. Barnes et al. (2006) have applied a similar peak matching method to electrical conductivity profiles in replicate EDC cores, with a correlation based automatic match routine, and have found 6600 tie points for a 44 ky record, or one tie point every 7 years on average. Our work suggest that the precision could reach one tie point every 1.5 years on average by analyzing the detailed chemistry record and potentially improving automatic matching routine to reach the accuracy of manual
matching. Concerning high resolution dating, while there will be no additional dating point to further refine the dating beyond volcanic horizons, as was done here with the 20 years snow pit dataset, we have showed that simple linear interpolation of the accumulation rate produces a timescale within the ± 1 year uncertainty. By combining 2 or more replicate cores at high resolution, this open the way of interpreting climatic signal in Dome C ice-core at higher frequencies, potentially reaching annual resolution.

## 5  Conclusions

Here, we reconstructed the last two decades of snow accumulation at Dome C, by using chemical composition and physical properties of 26 vertical profiles across a transect of 50 m. The dating method is based on using older snow pits to obtain absolute dates. The mean local accumulation rate matches the ones obtained in ERA5 (total precipitation - evaporation),



and from a stake farm (GLACIOCLIM, within the spatial uncertainty). The amplitude of the inter-annual variations of snow
accumulation can reach up to 50 % of the total local accumulation at Dome C.

We also evaluated the spatial variability of the accumulation at the meter scale. We found that variations of 20 % in the accumulation rate along the transect on average, with a decorrelation length of roughly 1.4 meters. These statistics confirm that snowfalls are not uniformly deposited at Dome C, but instead each event forms patches of accumulation. The measurements suggest around 10 % chances of annual accumulation hiatus at the profile scale. This impacts ice core signal interpretation and
contributes to the stratigraphic noise. We argue that for timescale larger than two years, the impact of these random erosion on the accumulation is largely mitigated, and that by combining several profiles, it is possible to evaluate precise local annual accumulation rates.

We also found that the holes in the local surface topography are filled only slightly more than the bumps (+15 %) at the annual scale) leading to persistence of the topography over many year. We indeed observed the persistence of a 30 cm dune
over the 20 years duration of the trench record. This shows that meter-scale topographic features may persist over decades of accumulation. Better understanding of the competing processes leading to the spatial pattern of the surface mass balance are key, as the detection of the influence of climate change on precipitation amounts in Polar Regions remains to be determined. In particular, dating ice cores which is often using volcanic eruptions only happening every several decades could suffer from uncertainties of a few years. Taking into account the accumulation patterns is thus key to reconstruct sub-decadal climatic
variations from ice core records in interior sites of the Antarctic Plateau.





# Appendix A

## A1    More details on the dating procedure

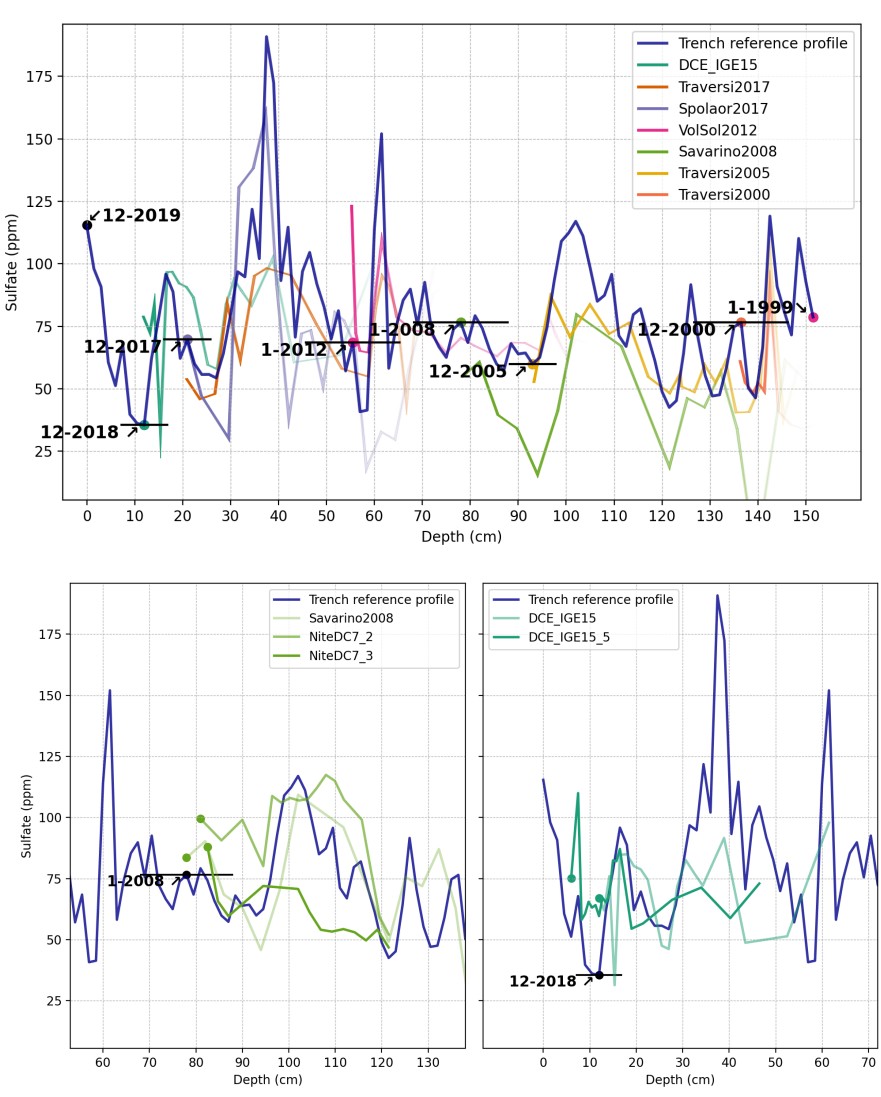

Figure A1: Illustration of the method for dating; the 6 snowpits profiles aligned with the reference. The fading on shows the first 30 cm for each profile. Bottom: two cases with duplicate snowpits for the sampling times. Different shapes for the top of the snowpits lead to variation on the corresponding depth on the reference profiles. The deviations are captured by the error bar





## A2 Surface snow density

In our results section, we are comparing accumulation timeseries obtained from the trench data, which takes into account the
density measurements in the trench, to accumulation timeseries obtained from RLS and stake farm measurements, computed
from elevation change, multiplied by a constant surface snow density.

This density is often taken as $320\ \mathrm{kgm^{-3}}$ in the litterature Genthon et al. (2016). We propose another value of $295\ \mathrm{kgm^{-3}}$
based on density measurements in the trench (Figure A2). While the density profile shows a very small increasing trend with
depth (slope = $11\ \mathrm{kgm^{-4}}$) that indeed has an intercept at $320\ \mathrm{kgm^{-3}}$, the mean density in the 0-6 $\mathrm{cm}$ depth near the surface,
which is the range where elevation change measurements take place, is only $297\ \mathrm{kgm^{-3}}$. While it cannot be excluded that
this is due to inter-annual variability of snow density (there is another density minimum at 30 $\mathrm{cm}$ depth), we observe that
reconstructed accumulation rates of RLS and GLACIOCLIM are more consistant with the trench with the lower surface snow
density value.



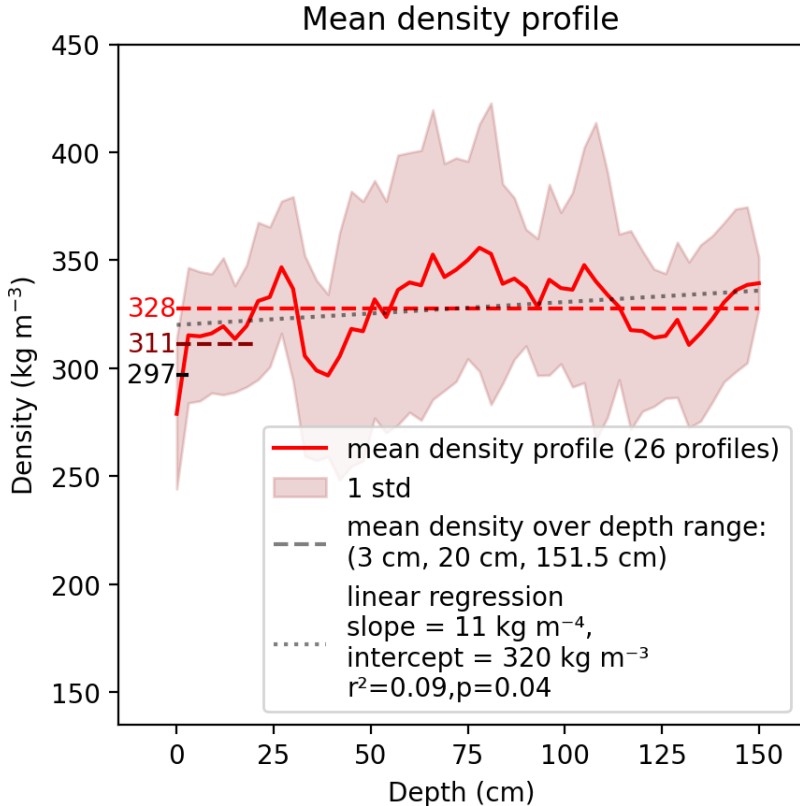

Figure A2: Mean density profile as a function of depth for the 26 evenly spaced out trench profiles. There is a high spatial variability, with only minor densification trend in the upper 1.5 m. We note that a linear regression gives a surface density of $320 \ \mathrm{kg \ m^{-3}}$ as in Genthon et al. (2016). However the surface snow (0-6 cm) in the trench is about 10% lower on average, at $295 \ \mathrm{kg \ m^{-3}}$. We compare both surface density values in the manuscript to reconstruct accumulation rates from stakes measurements. The stakes accumulation rates are more consistent with trench accumulation rates when using the lower surface snow density of $295 \ \mathrm{kg \ m^{-3}}$.



## A3 Supplementary figures for the evaluation of the alignment

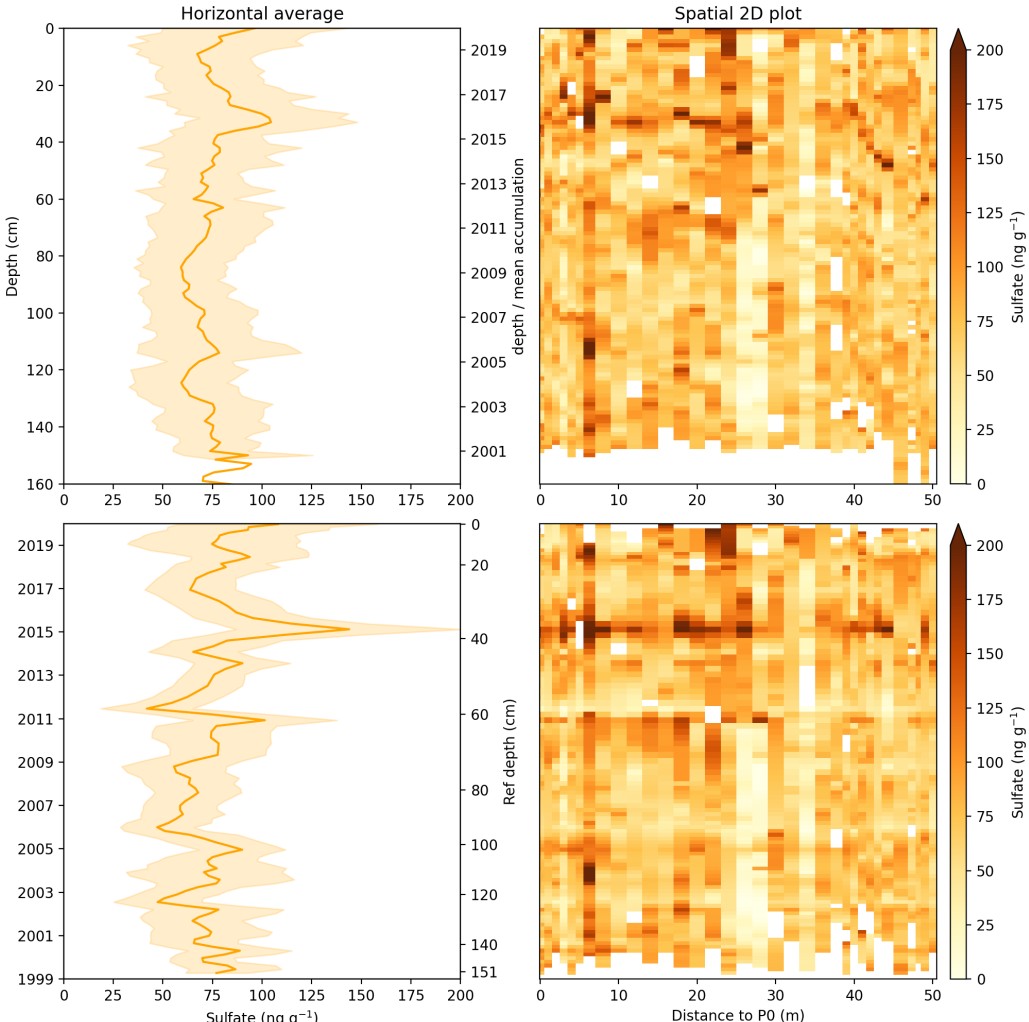

Figure A3: Effect of the alignment on horizontal averages of sulfate profiles in the trench. On top, sulfate profiles are given the simplest possible vertical alignment by setting the elevation of the surface to zero accross the entire trench. At the bottom, the common vertical scale is that resulting from the alignment. Envelope indicates $2\sigma/\sqrt{n}$ for the spatial standard deviation, while the solid curve represents the mean. Secondary $y$-axes on the right both figures show: on top the linear time-scale that would result from constant accumulation of $8\ \mathrm{cm\ yr^{-1}}$, at the bottom the depth scale is that of the reference profile (Sect. 2.2).





**A4   Sensitivity tests for dating and alignment**

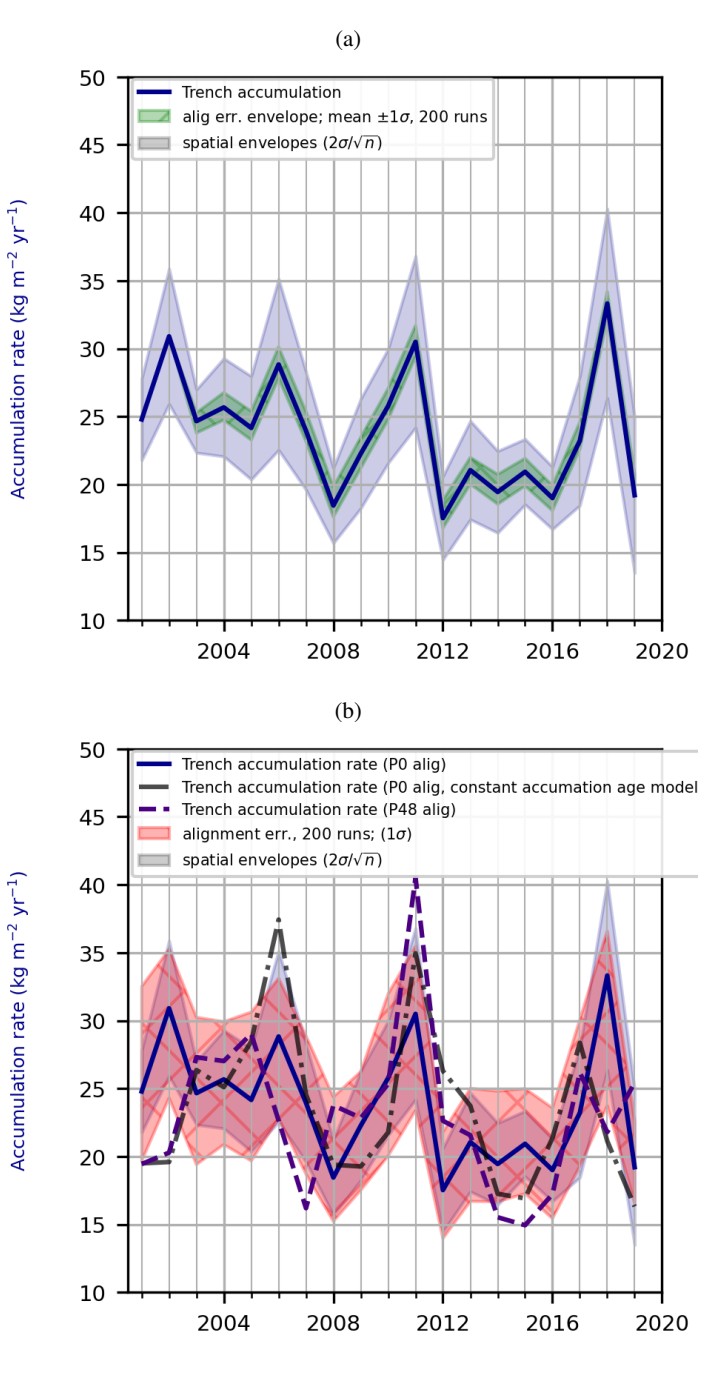

Figure A4





Figure A4: Sensitivity tests on the alignment and dating. [top], Montecarlo experiment with accumulation computation from isochrones modified by a uniform random variable of width $\pm\,3$ cm at each point (the alignment resolution) (200 runs). The envelope of all runs represents a standard deviation of 0.8 kg m$^{-2}$yr$^{-1}$. [bottom] Montecarlo experiment where the age model on the reference profile is changed at each point by a uniform random variable of width $\pm\,1$ year. (200 runs). The envelope of all runs represents a standard deviation of 5 kg m$^{-2}$ yr$^{-1}$. The dashed blue line represents the accumulation reconstructed from an independent alignment and dating performed with P48 as a reference profile. Although differences are not negligible, we see that they fall within the range of the dating sensitivity test around the P0 alignment. The dashed black line is the accumulation reconstructed with a linear dating on the reference. Again it is within the range of the sensitivity test.

## A5 Multi-annual accumulation dephasing

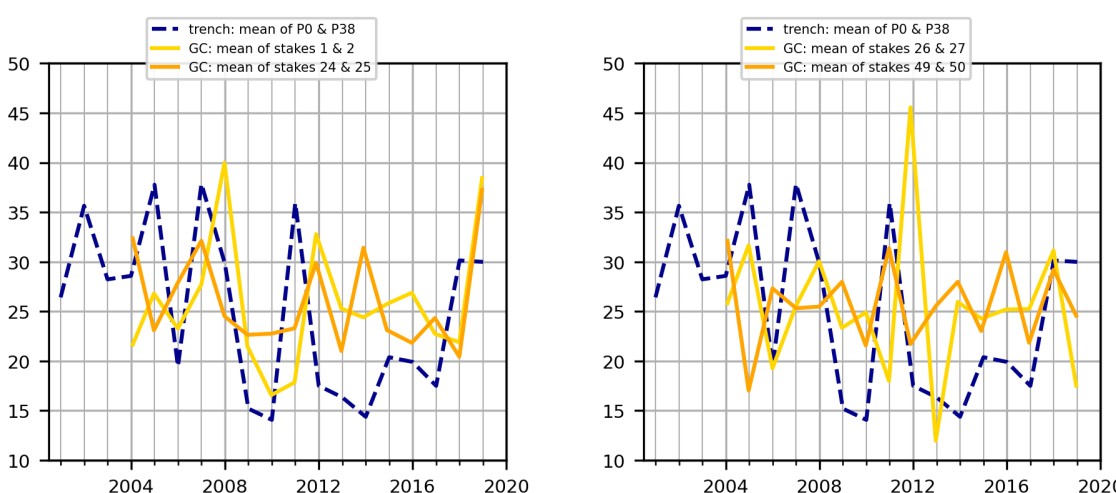

Figure A5: Mean accumulation time series of P0 et P38 compared against mean accumulation time series of four pairs of stakes approx. 40 m apart picked in glacioclim. We see that the phase mismatch of accumulation peaks is also present inside of the GLACIOCLIM stakes network itself, and we can find pairs of stakes (49 and 50) that have a maximum in 2011 just like the trench. Therefore there is a possibility that the phase mismatch in the mean accumulation time series of the paper is just a matter of spatial variability and contribution of km scale snowdrift on the snow accumulation.

## A6 Spatial decorrelation of sulfate

We adress the question of the decorrelation length of the sulfate signal. We want to extract a subset of profiles on which accumulation is free of neighbour to neighbour correlation of local noise. We compute the decorrelation length of the sulfate





signal in the aligned trench. We find an e-folding / decorrelation length of 1.3m (Fig. A6). Therefore, we choose to work on
the subset of 26 profiles that are evenly spaced out, located every 2 m inside the trench.

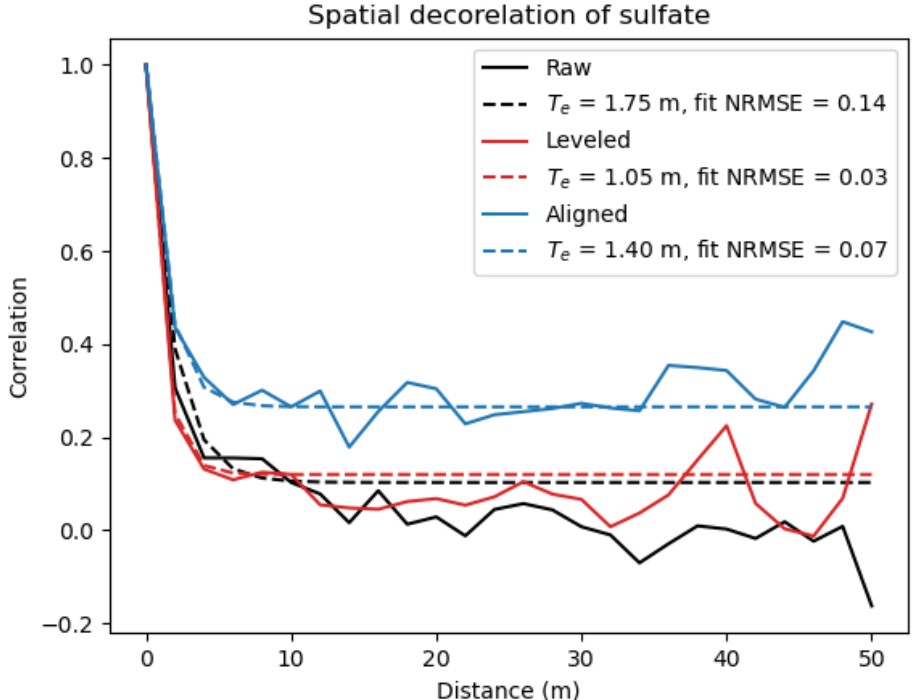

Figure A6: Spatial decorrelation of sulfate in the raw, leveled and aligned trench. We see the increase in mean value of inter-
profile correlation at long distance from 0.1 to 0.3 in the aligned trench. We also see the horizontal decrease in correlation with
1.3 m e-folding. This supports the choice of the 2 m pits subset for spatial diagnosis on alignment and snow accumulation
rates.

## A7    Computation of accumulation for the trench

To get snow accumulation time series from the dated isochrones we have obtained as a result of the alignment and dating of
the trench, we must convert elevation change computed from the height difference of the isochrones, to snow accumulation

rate. We could do this with a mean density value as in the case of the RLS and stake farm data. However, to capture spatial
variability as finely as possible, we exploit the 3 cm resolution density data available for all the profiles in the trench, and
compute accumulation rate based of the density integrated between two isochrones. An illustration of this computation, is
given in Fig. A7. We also provide a comparison with the accumulation rate based on a constant density.



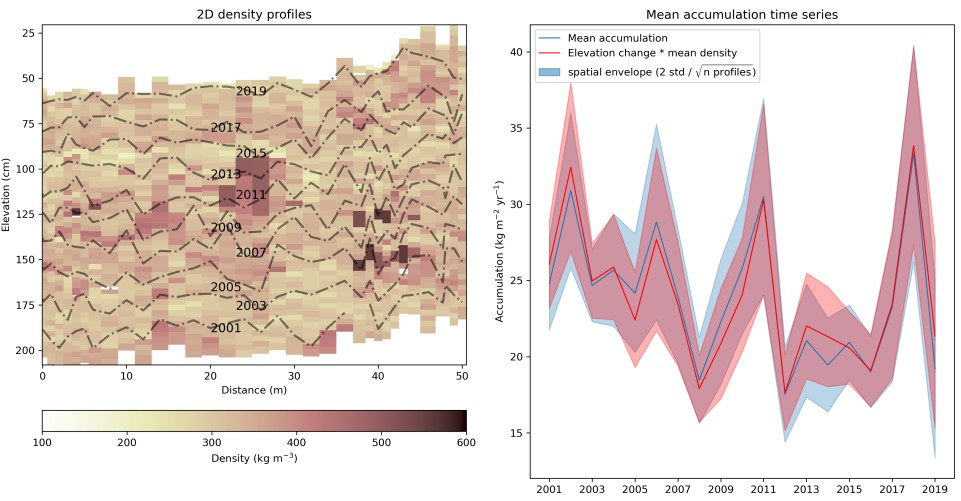

Figure A7: (left) Computation of the accumulation rate from the chemistry isochrones and the density measurement in the trench. Total snow accumulation is the density integrated between two isochrones. (right) Comparison of the accumulation time series obtained from the fully resolved density measurements and from a constant density of $328 \ \mathrm{kg \ m^{-3}}$. There is very little difference in the mean accumulation rates of the 26 profiles. This supports the use of a constant density for GLACIOCLIM stakes.

*Data availability.* The dataset will be submitted shortly to Pangaea and will be made available alongside the manuscript upon publication.

*Code and data availability.* The interface used for the alignment was coded using the python tkinter library for graphical interfaces. It was designed to be easy to install and to use, and applicable to any ice core data, from shallow snow pit to over 100 m deep cores. We wish that this code can support work from colleagues on similar ice / snow / firn core alignement challenges. It is available as a python package and can be installed as an executable python app on Windows, Mac or Linux distributions. See the reference page on the pypi.org database at https://pypi.org/project/lscealice/.

*Author contributions.* AO and MC prepared the manuscript with contributions from all co-authors.

*Competing interests.* The authors declare that they have no conflict of interest.



*Acknowledgements.* We acknowledge the French Polar Institute (IPEV) through the program 1110 (NIVO) and 1177 (CAPOXI 35–75). This research was made possible from funding from the DFG project CLIMAIC, the ERC Project SAMIR (HORIZON: European Research Council, grant no. 101116660), the Antarctic Science Bursary (project DATETRENCH), and the ANR project ANR-15-IDEX-02 for supporting data acquisition at Concordia station and laboratory analysis.




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
