# Peer review of "Inter-annual snow accumulation and meter-scale variability from trench measurements at Dome C, Antarctica"

_EGUsphere, 2025_

## Author Comment (AC1)

EGUSPHERE-2025-3259

Final response, December 18th 2025

Final author comments

**Response to RC1**

Ooms et al. reconstructed the snow accumulation time series at Dome C over the last 20 years, using chemical composition and physical properties from 35 vertical profiles in a 50 m long snow trench. This is a rare, high-resolution multi-profile trench dataset from the Antarctic Plateau, providing a reference for subsequent SMB research. Their results showed that the annual SMB broadly agrees with stake farm and ERA5, exhibits meter-scale spatial variability, and reflects persistent meter-scale topographic features over multiyear timescales. I believe this work will interest researchers studying surface mass balance (SMB), stratigraphic processes, and ice core interpretation on the Antarctic Plateau. However, the manuscript needs more confirmation and clarification before acceptance.

Major comments:

1. The key result of this study is the establishment of a high-resolution dating method for snow pits. The dating is achieved by aligning the target profile with the sulfate records of old snow pits. The process of manually selecting tie points for alignment is subjective. The article does not quantify the agreement between different operators.

We fully recognize the existence of operator dependent variability in the construction of the trench age model. The detailed alignment protocol (Section 2.2 line 120) is meant to guide the operator as much as possible and to reduce subjective choices.

We had performed replicability tests with other users which have been included in the revised version of the manuscript, in section A4, figure A4b. The results are very similar to sensitivity tests with the same operator using a different reference profile (P48) as a target for the alignment. The resulting accumulation estimate falls within the 20% uncertainty envelope of the global dating uncertainty.

In the main text, we have added the following to Section 4.2, line 352-372:

> "First, the reference profile can itself be subject to hiatus. In section 3.4, we found that annual accumulation hiatus occurs in 5% of the profiles on average, and 6 monthly hiatus about 40% of the time. Comparing the sulfate records in five 100 m long ice cores, Gautier et al. (2016) found a higher probability of 30% of missing one single volcanic event in a single core, which is consistent considering the peak deposition takes place over less than a year (Cole-Dai et al. 1999). Accumulation hiatus on the reference profile lead to missing minima or maxima in its sulfate record. Such a missing feature would be present in 95% of the trench profiles and inaccurately aligned on the reference. However, with our approach, these effects are automatically corrected for on average. For instance, if year 2015 was missing in the reference, but years 2014 and 2016 were clearly identified, the accumulation for 2014-2015 and 2015-2016 may be inaccurate, but the accumulation of the two years period 2014 2016 would still be correct. Sensitivity tests on the choice of reference profile, using P48 instead of P0 as an alignment target, indeed show such variations, changing the intensity or phasing of inter-annual accumulation, with a standard deviation of 22% (Sect. A4). Similarly, user induced variability when performing the alignment with the same reference P0, which will inevitably impact assignment of ambiguous features, is contained within the global uncertainties of the accumulation reconstruction, with a 20% standard

deviation on inter-annual accumulation values. We have considered the alternative of using a stacked profile as the reference to get a more representative average sulfate profile, but precisely because of stratigraphic noise, this produces a very smooth profile (Appendix, Figure A3.a) on which it becomes impossible to recognize any feature at the inter-annual scale."

2. I believe that all readers will want to know why this study uses a single profile P0 as the reference to align all other sections in the entire 50-meter trench.

(1) If P0 itself is abnormal, then this anomaly may be passed on to all other profiles, causing a deviation in the dating of the entire [trench]. Münch et al. (2016, https://doi.org/10.5194/cp-12-1565-2016) pointed out it is necessary to average multiple ice cores or snow pits to extract climate signals and filter out stratigraphic noise.

The choice of using a raw, non-stacked reference profile is indeed a very complicated topic, and is a deliberate choice given the constraints of our alignment and dating method. Figure A3.a in the appendix illustrates the fact that averaging out profiles without alignment produces a very smooth profile. It becomes difficult to compare this stack with the individual profiles of the trench. Our goal here was to trace isochrones in the trench by exceptional markers that leave non uniform signature in the different profiles, hence the deliberate choice to work with a raw profile as a reference, which will contain some, but not all, of these signatures.

The second step of dating the reference profile is exactly meant to reduce the bias caused by anomalies of the reference profile, since periods with more or less accumulation, and accumulation hiatus, are squeeze and stretched accordingly. The method is limited by the little amount of historical snow pits available, and comes with large uncertainties, but it is our best attempt to reduce the impact of using a raw imperfect reference profile.

These questions are already addressed in the discussion section of the manuscript (Section 4.2). We added a reference to this section when introducing the reference profile in the method to guide the reader (line 124):

> "We chose to use the single raw profile P0 as the reference profile because of its highest resolution (1.5 cm instead of 3 cm) and the presence of several sulfate features that are suitable as an alignment target. Despite these advantages, we observed that P0 features a lower SSA in its top snow than other profiles, and that on some profiles up to 3 cm of upper snow could not be matched to this reference. We interpret these facts as a missing snow layer at the top of P0. To solve this issue, we completed the 3 cm upper part of P0 with the mean surface
> snow properties in the 3 cm upper part of all profiles over the trench. This permits us to align all profiles with respect to the reference up to the surface (Supplements, Sect. S2.1). Similarly to this missing snow layer at the surface, in any single raw profile, there will be missing layers that can hinder alignment with other profiles. Yet, we chose to stick with aligning against a single reference profile because a pre-averaged profile is too smooth to obtain confidence in identifying successive peaks. We realized sensitivity tests about the impact of aligning on a single reference profile, as described in the Discussion section 4.2."

In the discussion section, we included the following additions (line 353):

> [last sentence in dark red in the text addition mentioned in point 1.]

(2) When P0 is compared with snow pits that may be hundreds of meters or even farther away, the basic assumption is that the sulfate signal is stable over a considerable spatial scale. However, an important finding of this study is that there is a significant meter-scale spatial variability in snow accumulation. Gautier et al. (2016, https://doi.org/10.5194/cp-12-103-2016) indicates that even ice cores just a few meters apart may have missing records of volcanic events.

We decided to compare raw profiles, keeping in mind that feature identification will fail for a certain fraction of snow pits due to the anomalies in the reference. This actually lead to only 13 snow pit profiles being used in the end because we could not identify the peaks corresponding to the upper 15 cm of the snow pit profile with the reference profile. Here, an important point comes from the fact that we do not need to include all the events, but enough to find common feature to add tie points to our reference profile.

However the point was not made explicitly that this 41% (9/22) discarding rate can be attributed to hiatus, inline with the observations of Gautier et al. They mention that, for a selection of two cores, there is a 60% chance of observing a volcanic peak in the two cores. Consequently, if we transpose our method, i.e. using one core to date a reference core, this attribution would fail for 40% of peaks. The numbers are not directly comparable, as Gautier uses volcanic peaks above +2 sigma, while our features are in the +1-2 sigma range, but the probabilities are of the same order.

Consequently, we have added the explicit reference to the Gautier study on line 140:

> "From the set of 22 snow pits dug at Dome C in the last 20 years (Gautier et al., 2016; Traversi et al., 2009; Caiazzo et al., 2021) we selected 13 that could be aligned with the reference profile of the trench (the others showed no remarkable feature in common with the reference in their upper 15 cm section). This 41% discarding ratio is comparable to the observations of Gautier et al. 2016, where they showed that a volcanic peak only has a 60% chance of being present in any two given cores."

Additionally, just like in the case of trench inter-profile matching, a stacked reference profile would be difficult with the historical snow pits, whether or not they are replicated themselves. For instance, the VolSol cores from the study of Gautier 2016 are only replicated beyond 5 m depth, so that only one profile can be used for our applications. Since our goal here was to identify with the highest level of certainty how to align our trench with historical profiles, it is expected that there will be hiatuses both in our reference profile and in the other profile we're using for dating. We acknowledge that this point was not made clearly and added the following paragraph in section 4.2 to clarify our approach:

> "We have chosen to work with a reference profile as a target for the alignment of all the profiles in the trench, instead of pair by pair alignment. Beyond the obvious gain in time in the alignment ($n$ vs $n(n − 1)/2$ pairs to align, where $n = 35$ is the number of profiles) and dating (only one profile to date), we also gain in coherence as all features recognizable on the reference were matched precisely.
> For the same reason, we chose to work with a single raw reference profile instead of a trench stack. Indeed, the older snow pits themselves are mostly single, non-replicated profiles. Stacking the trench unaligned profile produces a smooth profile (Appendix, Fig A3.a) which becomes unpractical for peak matching with older snow pits. By using a single raw profile, we were unable to match 9 out of the 22 snow pits used for dating (due to the hiatus or gaps in the reference and in the snow pits themselves), but we keep a high confidence for successful identification of the remaining 13 snow pits. This 60% ratio is consistent with the two ice-core volcanic peak identification rate of Gautier et al 2016. Reference profile

alignment however comes with some caveats, and the dating of the reference remains the main source of uncertainty."

(3) The manuscript indicates in the method section that due to too little snow was available to measure chemical elements of the P0 sample, it was mixed with a high-resolution sample from another profile. The authors did not explain the specific method of mixing, and this operation may artificially change the chemical signal of the reference profile.

We acknowledge that the method of mixing was not clearly described in the method. The fact that it changes little to the composition of the sulfate profile is supported by the decorrelation length of 1 m. We have added the following sentence in the relevant paragraph:

At the localization of the first profile (P0), we actually realized four profiles (P-0.3, 30 cm before the reference profile, P0.1, 10 cm after the reference profile, and P0.3, 30 cm after the reference profile. Because the decorrelation length is roughly 1m (Appendix, Fig. A6), it is possible to mix snow profiles 10cm and 30cm away, respectively. Here, some of the snow samples of P0 did not have sufficient amount of snow to realize the entire span of chemical measurements (anions and cations), so in this case, we used extra snow from P-0.3 to complement. "

So I am skeptical about using a single and mixed profile P0 as the reference for the age alignment of the 50 m trench.

Let us summarize our answer to the three points above: We are aware that a single reference profile is subject to stratigraphic noise. Our point was not to have a perfect reference profile, but something easy to work with and that proves to work very well both for inter-profile alignment (1) and for older snow pit alignment (2) when common features are recognized in matching profiles (~60% of the time). Furthermore, we know by the observed spatial decorrelation length of sulfate that the fact that P0 is a mix of two profile has little impact on the signal.

The uncertainties associated to using a raw reference profile as a target for the alignment are already presented in the manuscript (Section 4.2, line 360). But we had omitted to state clearly in the main text that this uncertainty by itself contains the variations seen in sensitivity tests, when using the profile P48 or when alignment is performed by another user (Section A4, figure A3b).

Therefore we have added the following to Section 4.2, line 352-372:

[Same section as the updated section mentioned in the first point, concerning user dependent variability].

3. The authors take the mid-2015 isochrone as an example to validate past topography. Using imperfect individual cases to prove isochrony over a twenty-year period is too weak. I suggest adding more isochrone for verification rather than relying on a single "perfect case". For example, Sinnl et al. (2022,  https://doi.org/10.5194/cp-18-1125-2022 ) emphasize the use of multiple time periods and various indicators to validate age model.

We agree that it would be better to have more isochrones to compare the trench alignment with. Unfortunately, we did not identify any other physical isochrones that could have been linked with a time marker in the rest of the trench.

Another indication that gives us confidence in the age model comes from water isotopes analysis, which are part of a manuscript being written, in which it is possible to identify two layers in 2014

and 2015 which can be recognized with the surface snow isotopic composition in d18O and d-excess. We see a strongly enriched d18O layer which our age model attributes to 2014, and a depleted d-excess layer attributed to 2015. Comparable modification in the isotopic composition of surface snow for these periods have been measured and explained by the influence of sublimation flux and in firn vapor exchange for these two exceptional years (Casado et al. 2021). Unfortunately, this manuscript hasn't been submitted yet, so it is difficult to include this in the present study. More importantly, because these two other dates are so close to the one we were able to identify in Mid-2015, it is not adding any constrain on the dating uncertainty.

In addition, an enriched d18O layer in 2002-2003 provides another potential stratigraphic marker, which is not covered by surface snow observations, but which we are in the process of modeling with isotope enabled GCM models. The isotope stratigraphy is outside of the scope of the present study, but it will come as an independent validation of the chemical and physical stratigraphy.

[Figure]

(a) (b)

Additional Figure 1: stable water isotopic composition of the trench, with d18O (left) and d-excess (right)

Instead, the validation of the age model at +/- 1 year is based on the little divergence of snow pit age models (obtained by linear interpolation to the Pinatubo horizon, transferred to the reference profile) as shown in figure 5. All models converge toward early 1999 for 150 cm depth. We recognize that this was not stressed enough in the manuscript and added the following sentence for emphasis on line 216:

> The next six dating points were obtained by alignment of the reference profile to snow pits dug in the Dome C area over the past 20 years. The depth uncertainty of dating points (horizontal error bars in Fig. 5) leads to an uncertainty in the snow age at a given reference depth (vertical error bars) ranging from ± 0.5 years for the January 2019 tie point to ± 1.5 years for the January 2012 tie point. The average limited spread of the individual age models (doted colored lines in Fig. 5) has a mean standard deviation of 0.7 years between models and gives confidence in the accuracy of the trench age model at ± 1 years.
>
> The last tie point at the bottom of the reference profile is obtained from the mean over the six linear age models transferred from the aligned snow pits reaching the Pinatubo horizon (Table 1). The standard deviation among the age models gives an
> error estimate. They indicate that the snow layer buried at 151.5 cm depth was deposited in early 1999 ± 1 years. From the surface snow deposited in June 2019, the trench thus archives about 21 years of snow accumulation, with a mean elevation increase of 7.3 cm yr$^{-1}$ .

We also want to emphasize that, while validating the stratigraphy at the inter-annual scale is important, it is something which will be out of reach for many ice-core applications where no

previous ice-core are available. The more important point we want to make is that a linear age model (gray line in Figure 5) gives annual accumulation time series (Figure A3(b) dashed black) that sits within the dating uncertainty (Figure A3.b red envelope). In other words, we argue that linear age model spanning a few decades (such as between two volcanic horizon) coupled with the alignment of replicate cores will give a good estimate of inter-annual accumulation.

To summarize, while we are not able to provide more isochrone for verification, the validation given by the 2015 isochrone was just given as a striking example showing that linear interpolation of the age model on a 5 year period can fall very close to a given time marker.

Minor comments:

1. The horizontal decorrelation length of the sulfate signal is 1.26 m in the aligned trench in L204. However, in Figure A6 of the appendix, the legend shows the aligned decorrelation length as "1.40 m".

Thank you for pointing this out. This has been fixed to "1.40m" in the text, as in the figure.

2. Only 6 dating points were kept for the age model in L144, but table 1 lists seven different records of snow pits/ice cores.

The table lists explicitly the two snow pits for 2017 because they both have a linear age model based on volcanic horizon, which are useful to validate the age model. Naming them makes Figure 5 more readable. The caption of the table was updated as follow to remove the ambiguity:

> "Summary information for the snow pits and ice cores  used in the dating.  Last column indicates whether the Pinatubo volcanic horizon of 1992 can be identified in the sulfate signal, allowing us to compute a linear age model for that snow pit. Only the snow pits whose upper tie-points were used for dating are shown (non duplicates, 6 out of 13 snow pits), except for Traversi2017, which is redundant, but whose linear volcanic age model is shown alongside other snow pits in Fig. 5 for validation."

L24: "offsetted" should be "offset"

This has been fixed.

L38/57: unify spelling of "snow pit / snow pit".

All instances have been spelled as "snow pit"

L73: Could you explain the direction of the dominant wind?
The paragraph has been updated to:
> "The trench was perpendicular to the main orientation of the wind and the sampling was carried out on the wall face sheltered from the wind, which is southerly on average at Dome C (Genthon et al., 2021)"

L78: remove height .
This has been fixed.

L81: "was" should be "were".
This has been fixed.

Figure 1 caption it is more appropriate to change the "+" to "follow by".
This has been fixed.

Figure 1 caption: "1.5cm" should be "1.5 cm".
This has been fixed.

L116: Could you tell me why you chose SSA > 40 as the threshold?

This value is based on observations of Libois et al. 2015 at Dome C over the course of two summer seasons (2012-2013,2013-2014) where they show that precipitation events always have a signature in increase of surface snow SSA to values higher than 40 m^2/kg, and that when there is no precipitation, surface snow SSA will decrease to values under 40 m^2/kg over the course of only several days. Therefore we use this threshold to identify snow necessarily linked to recent precipitation events.

To further strengthen this choice of threshold value, we have now added a sensitivity test to the manuscript (Supplement S1). We reproduce the figure here for convenience:

[Figure]

The following sentence was added to the text:

> As the trench was sampled at the beginning of summer, before significant metamorphism had occurred (Picard et al., 2012), we used the high SSA values ( > 40 m2 kg−1 ) of recently deposited winter snow (Libois et al., 2015) to match the topmost portions of the profiles and to detect hiatus in snow deposition. We have performed a sensitivity test to ensure that this was also an appropriate threshold for the trench dataset. We counted all SSA values in the trench dataset (26 evenly distributed profiles) above a certain threshold, for threshold values ranging from 20 to 60. We see a clear transition around 38-42 m2 kg−1, where the data point count increases sharply under 38 m2 kg−1, indicating a longer persistence of such SSA values during grain coarsening. This confirms that 40 m2 kg−1 is a sweet spot to identify fresher snow. The details and corresponding figures are provided in supplement S1. The rest of the profile is aligned by matching sulfate peaks, first with the largest peaks and then refined with sub-features.

L175: Does this indicate that the previous use of 320 as the density value of Dome C is too high?
This is indeed what our accumulation reconstruction seems to indicate. The density analysis carried out on the trench data (Figure A2) shows the same bias: a linear interpolation on 1.5m of snow at 3 cm resolution, as was done by Genthon et al. 2016 on 1m of snow with 10 cm resolution, gives a

surface density of 320 kg/m2. The observed density however in the upper section of the trench is closer to 300 kg/m2. This could be explained by grain coarsening due to snow metamorphism during the first year of residency. Consequently, fresh snow readings on the snow stakes should be converted to mass using a lower density.

L487: "alignment" should be "alignment"
This has been fixed.

---

## Author Comment (AC2)

EGUSPHERE-2025-3259

Final response, December 18th 2025

Final author comments

**Response to RC2**

The manuscript presents work in the area in Antarctica with very low accumulation and mainly no ablation. That gives excellent possibility to study ages of different snow layers. Aim is to define how well annual stratigraphy is visible in older snow and how surface features affect to the stratigraphy over the years. In addition, SMP measurements (snow hardness) correlation with age defined based on chemical composition is studied. Also, annual accumulation rates calculated from the trench, ERA5 and earlier snow stake data are compared. For the analysis, 50m long and 1.5m deep trench was made in December 2019, covering snow accumulation approximately from past 20 years.

**General comments**

Accumulation hiatus could be explained better.

Accumulation hiatus, near zero accumulation or erosion over a given period of time, is a concept that has been studied already, see for instance these manuscripts (Genthon et al., 2016, Picard et al. 2019). We add this definition to paragraph 3.4 as well as the link to these manuscripts to ensure that readers have the same definition than the one we used when they read this manuscript.

The novelty of our method is to describe accumulation hiatus from a final accumulation product (snow pits) in which, by definition, only strictly positive accumulation is recorded. To do so we have extended the definition of hiatus to a snow accumulation less than 3 cm over a given period of time. 3 cm correspond to the smallest accumulation amount we can detect with the trench resolution.
Next we do a spatial analysis of accumulation hiatus based on the dated trench isochrones covering a 50 m distance and a 20 year accumulation record. The vertical distance between two isochrones represents the accumulated snow over that period at the location of the 26 trench profiles. We ask what fraction of profiles are subject to accumulation less than 3 cm of snow for that period of time. Finally, we repeat this computation for all pairs of isochrones, and bin average the results by time period, to obtain the spatial fraction of hiatus as a function of time period. We find that the trench isochrones reproduce well the hiatus that are observed by direct method such as with the RLS.

The have rewritten the first paragraph in the hiatus section 3.4 in order to improve the explanation.

Old version:

> "Fig. 4b shows that the 2012 and 2013 isochrones are in a very narrow range of depths (< 3 cm ) around profile P32, indicating a low accumulation at this location for the year 2012. Such occurrences of accumulation below 3 cm are interpreted as near accumulation hiatus (compared to the 8 cm of annual snow accumulation), i.e. periods with negligible accumulation or even erosion at a given location. Considering that the vertical resolution of the alignment is 3 cm, i.e. the lowest depth difference that we can resolve, we refer to periods with smaller amount of snow accumulation as *hiatus*. The hiatus fraction for an

accumulation event is the fraction of profiles with a hiatus. The probability of hiatus for a time scale is the mean hiatus fraction for all accumulation events corresponding to this time scale. The frequency of hiatus should be close to 100 % for a very short time difference, because almost all snow deposits are less than 3 cm, and gradually decreases with the considered time-scale.

Fig. 7 shows the probability of hiatus occurrence as a function of the time scale, up to three year periods, in the trench and in the RLS area. The 19 year record of accumulation in the trench, with 3 cm vertical resolution, equivalent to at least 6-month temporal resolution, is compared to the 2 years time series of the RLS (period 2) with daily resolution. (...)"

New version:

"Annual accumulation hiatus, near zero accumulation or erosion over a year, is frequently observed near Dome C (Genthon et al., 2016, Picard et al. 2019). We investigate whether we can detect it in the reconstructed trench accumulation. In order to describe accumulation hiatus from a final accumulation product (snow pits) in which, by definition, only strictly positive accumulation is recorded, we have extended the definition of hiatus to a snow accumulation less than 3 cm over a given period of time. This 3 cm threshold corresponds to the smallest accumulation amount we can detect with the trench resolution. For example, the 2012 and 2013 isochrones in Fig. 4b are in a very narrow range of depths (< 3 cm ) around profile P32, indicating an accumulation hiatus at this location for the year 2012. We consider all pairs of isochrones and compute the fraction of profiles where hiatus are occuring. We bin average this hiatus fraction by the time period separating isochrones in each pair. The same analysis is repeated on the elevation profiles of the RLS data, with a threshold of 3 cm accumulation.

The hiatus fraction as a function of time period for the trench and RLS is shown in Fig. 7. The 19 year record of accumulation in the trench, with 3 cm vertical resolution, equivalent to at least 6-month temporal resolution, is compared to the 2 years time series of the RLS (period 2) with daily resolution. (...)"

The main point and impact of the study could be better clarified in the conclusions.

We acknowledge that the conclusion can gain in conciseness, by highlighting clearly three main points of our study, which were slightly hidden in too vague sentences. The three points are the following:

1) The global consistency of the reconstructed accumulation rate at Dome C with direct observational methods, both in mean values are inter-annual variability, including periods of low accumulation (accumulation hiatus).
2) We highlighted the persistence of a 30 cm high and 10 m long snow dune over a period of 20 years, and further showed the low correlation between topography and snow accumulation. This is a single observation at the location of the trench that would need to be generalized, but for interior sites like Dome C, 30 cm corresponds to more than 3 years of accumulation, so it is significant that it would remain at the same location over an extended period of time.
3) Finally, our results show that the linear approximation of the age scale age scale over 20 years is within a +-1.5 years dating error of the finer age scale based on using older snow pits to obtain absolute dates. This is an encouraging perspective for transposing our approach to the broader study

of replicate cores covering longer time scales, where only linear interpolation between multi-decadal time horizons are available.

We have added a stronger focus on the main points and impacts in the conclusion:

"Here, we reconstructed the last two decades of snow accumulation at Dome C at annual time scales, by using chemical composition and physical properties of 26 vertical profiles across a transect of 50 m.  The mean local accumulation rate matches the ones obtained in ERA5 (total precipitation - evaporation), and from a stake farm (GLACIOCLIM, within the spatial uncertainty). The reconstructed accumulation time series shows further similarities with observations, including inter-annual variations that  can reach up to 50 % of the total local accumulation, and around 5-10 % chances of annual accumulation hiatus at the profile scale

We evaluated the spatial variability of the accumulation at the meter scale. We found that variations of 20 % in the accumulation rate along the transect on average, with a decorrelation length of roughly 1.4 meters. These statistics confirm that snowfalls are not uniformly deposited at Dome C, but instead each event forms patches of accumulation.  This impacts ice core signal interpretation and contributes to the stratigraphic noise. We argue that for timescale larger than two years, the impact of these random erosion on the accumulation is largely mitigated.

We observed the persistence of a 30 cm dune over the 20 years duration of the trench record. We also found that the holes in the local surface topography are filled only slightly more than the bumps (+15 % at the annual scale) leading to persistence of the topography over many year.  This shows that meter- scale topographic features may persist over decades of accumulation. Better understanding of the competing processes leading to the spatial pattern of the surface mass balance are key, as the detection of the influence of climate change on precipitation amounts in Polar Regions remains to be determined. In particular, dating ice cores which is often using volcanic eruptions only happening every several decades could suffer from uncertainties of a few years.  Yet, our results show that a the linear approximation of the age scale between tie points separated by a few decades is only leading to uncertainties of 1.5 years. This is an encouraging perspective for transposing our approach to the broader study of replicate cores covering longer time scales, where only linear interpolation between multi-decadal time horizons are available."

Please write numbers below 10 as text.

This has been corrected wherever necessary.
Numbers were left as numerals when preceding a unit or amount of time (2 meters , 6 months, 5 years), when indicating a version (version 2).

Please avoid starting sentences with "Figure …"

This has been fixed.

I recommend using the CRediT standard for the author contributions. Now it is not clear if any of the authors participated in the field measurements, for example.

Details on the contribution of all authors are now provided using the CRediT standard.

**Specific comments**

Figure 1: P0.3 is not marked to the Fig 1a.

This was the result of an inconcistency when mentioning profiles P0.1 and P0.3 whose samples were mixed together. We are now only mentioning P0.3 in the text and showing P0.3 in Fig 1a.

Line 87: Why P0 was not mixed with P0.1 since it is closer than P0.3, right?

We wished to have two complete profiles at a short distance of each other. None of P0, P0.1, P0.3 had big enough snow samples for the analysis. Our choice therefore was to group P0 with a profile 30cm to the left ("P-0.3", not mentioned in the manuscript) and group P0.1 and P0.3 together. Grouping P0.1 with P0 would have left P0.3 incomplete.

Line 63: Change as "Chemical tracers and physical snow properties including snow microstructure (density, specific surface area (SSA) and penetrometry)".

This has been fixed.

Line 64:  Change as "methanesulfonic acid (MSA)"

This has been fixed.

Line 164: Change as "The stakes are measured every year"

This has been fixed.

Figure 5: Colors are difficult to separate and makes figure difficult to read.

The colors in Figure 5 are only meant as a aestetical device and are not essential to the reading of the information in the plot. This is most easily seen by looking at the picture in gray scale (see below). What is interesting is to look at the overall spread of the snow pit age scale, rather than knowing which age scale corresponds to what snow pit, which is the information contained in the color. Overall, we see that the snow pit age scales are well included into the error bars and that they converge towards the same value +/- 1 year at the bottom of the reference profile (corresponding to early 1999).
This is the very reason why colors where chosen similar enough so the figure does not appear

saturated, which would distract the reader, but not completely homogeneous so as to slightly separate the profiles.

[Figure]

Figure 6b: Figure is very unclear because all lines are top of each other.

Figure 6b is the integration of the accumulation rate time series shown in Figure 6a. The point is exactly to show that they show very little deviations and reach the same values after 15 years of accumulation (2004-2019), so the difference in inter-annual variability seen in Figure 6a actually averages out when looking at total accumulation. As such, it is a validation of the trench accumulation rate reconstruction compared against reanalysis and stake measurements.
It also shows that the stake farm measurements with the classical surface density conversion (320 kg /m3) deviates towards higher total accumulation, while the suggested conversion value of 295 kg reconciles direct observations with the trench analysis and the ERA5 reanalysis.
The alternative of showing the 4 curves on shifted levels would fail to illustrate this important conclusion.

We have detailed the following sentence in the caption of Figure 6b to insist on this point:

> [...] (b) Cumulated accumulation time series. Series have been set to an initial value of 0 mm in January 2004 (first GLACIOCLIM stakes measurements), and RLS initial value is set to match GLACIOCLIM in January 2015. . When overlaid, the accumulation time-series of the trench follows closely that of ERA5 (precipitation - evaporation), and that of GLACIOCLIM with ρsurf = 295 kg m−3, showing a small deviation of less than 20 kg/m^2 (5%) over the period 2004-2019.